# DuetGraph: Coarse-to-Fine Knowledge Graph Reasoning with Dual-Pathway Global-Local Fusion

**Jin Li**[1,3,†], **Zezhong Ding**[2,3,†], **Xike Xie**[1,3*]

[1]School of Biomedical Engineering, University of Science and Technology of China (USTC)
[2]School of Artificial Intelligence and Data Science, USTC
[3]Data Darkness Lab, Suzhou Institute for Advanced Research, USTC
{lijinstu,zezhongding}@mail.ustc.edu.cn, xkxie@ustc.edu.cn

## Abstract

Knowledge graphs (KGs) are vital for enabling knowledge reasoning across various domains. Recent KG reasoning methods that integrate both global and local information have achieved promising results. However, existing methods often suffer from score over-smoothing, which blurs the distinction between correct and incorrect answers and hinders reasoning effectiveness. To address this, we propose DuetGraph, a *coarse-to-fine* KG reasoning mechanism with *dual-pathway* global-local fusion. DuetGraph tackles over-smoothing by segregating—rather than stacking—the processing of local (via message passing) and global (via attention) information into two distinct pathways, preventing mutual interference and preserving representational discrimination. In addition, DuetGraph introduces a *coarse-to-fine* optimization, which partitions entities into high- and low-score subsets. This strategy narrows the candidate space and sharpens the score gap between the two subsets, which alleviates over-smoothing and enhances inference quality. Extensive experiments on various datasets demonstrate that DuetGraph achieves state-of-the-art (SOTA) performance, with up to an **8.7%** improvement in reasoning quality and a **1.8×** acceleration in training efficiency. Our code is available at https://github.com/USTC-DataDarknessLab/DuetGraph.git.

## 1 Introduction

Knowledge graphs (KGs) are structured representations of real-world entities and their relationships, widely applied in domains such as information retrieval [1, 2], logical reasoning [3, 4], recommendation systems [5, 6], materials science [7, 8], and biomedical research [9, 10]. However, existing KGs are often incomplete, missing certain factual information [11, 12], which limits their effectiveness in downstream applications. As a result, inferring and completing missing entity information through KG reasoning is essential.

KG reasoning faces two fundamental challenges. Firstly, it requires effective aggregation and propagation of local neighborhood information to capture multi-hop and subgraph patterns among entities. Secondly, it must capture global structure and long-range dependencies across large-scale graphs to understand complex relationships that span multiple intermediate nodes. To address these challenges, a substantial line of previous research has been dedicated to developing methods to capture local neighborhood and global structure for KG reasoning. These methods can be categorized into two types: *message passing-based methods* and *transformer-based methods*.

Message passing-based KG reasoning methods [13, 14, 5] effectively capture local structural information via message passing mechanism [15], but often fail to model long-range dependencies and

---

*Corresponding Author    †Equal Contribution

global structural patterns [16, 17]. In contrast, transformer-based KG reasoning methods excel at capturing global KG information and long-range dependencies but tend to overlook important local structures or short-range dependencies between neighboring entities [18]. To address these limitations, recent state-of-the-art (SOTA) studies [19, 20] have shifted to integrate both local and global information by stacking message-passing networks and attention layers in a single stage.

However, such a single stage stacking approach tends to result in the problem of score **over-smoothing**, where incorrect answers receive scores similar to correct ones (Figure 1[2]), making them hard to distinguish. Accordingly, we summarize the problem into two core challenges. **Challenge 1:** Existing studies [23, 24] have shown that stacking either message-passing or attention layers individually deepens information propagation and aggravates the over-smoothing problem. When message-passing and attention layers are stacked together, these effects accumulate. **Challenge 2:** The discriminative capacity of single stage models is typically limited [25], as they generate the answer directly based on a one-shot reasoning. This deficiency in discrimination further exacerbates the over-smoothing phenomenon [26].

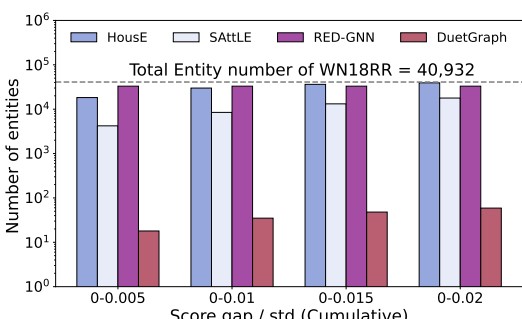

Figure 1: **Discriminative ability of KG reasoning models**: HousE [21], SAttLE [22], and RED-GNN [5] show limited discrimination, with many incorrect answers scoring close to the correct one. In contrast, our DuetGraph achieves clearer score separation, with far fewer incorrect answers near the correct score.

To address these challenges, we propose **DuetGraph**, a coarse-to-fine KG reasoning mechanism with dual-pathway global-local fusion. To address **Challenge 1**, we propose a dual-pathway fusion model (Section 3.1) that separately processes global and local information before adaptively fusing them. By segregating message-passing and attention layers, rather than stacking them, our model alleviates over-smoothing and improves reasoning quality. For **Challenge 2**, we propose the coarse-to-fine reasoning optimization (Section 3.2), which first employs a coarse model to predict and partition candidate entities into high- and low-score subsets, and then applies a fine model to predict the final answer based on the subsets. It enhances robustness against over-smoothing and improves reasoning quality. We theoretically demonstrate the effectiveness of coarse-to-fine optimization by mitigating over-smoothing in Section 3.2. Furthermore, we demonstrate the effectiveness of this optimization in improving inference in Section 4.4.

Our contributions can be summarized as follows. **1)** We propose DuetGraph, a novel KG reasoning framework to alleviate score over-smoothing in KG reasoning. Specifically, DuetGraph: a) utilizes a dual-pathway fusion of local and global information instead of a single-pathway method, and b) adopts a coarse-to-fine design rather than one stage design. **2)** We theoretically demonstrate that our proposed dual-pathway reasoning model and coarse-to-fine optimization can both alleviate over-smoothing, thus effectively enhancing inference quality. **3)** DuetGraph achieves SOTA performance on both inductive and transductive KG reasoning tasks, with up to an **8.7%** improvement in quality and a **1.8×** acceleration in training efficiency.

## 2 Background

**Knowledge Graph.** A knowledge graph (KG) is a structured representation of information where entities are represented as nodes, and the relationships between these entities are represented as edges. Typically, a KG $\mathcal{G} = \{\mathcal{V}, \mathcal{E}, \mathcal{R}\}$ is composed of: a set of entities $\mathcal{V}$, a set of relations $\mathcal{R}$, and a set of triplets $\mathcal{E} = \{(h_i, r_i, t_i) \mid h_i, t_i \in \mathcal{V}, r_i \in \mathcal{R}\}$, where each triplet represents a directed edge $h_i \xrightarrow{r_i} t_i$ between a head entity $h_i$ and a tail entity $t_i$.

**Knowledge Graph Completion.** Given a KG $\mathcal{G} = (\mathcal{V}, \mathcal{E}, \mathcal{R})$, KG completion is to infer and predict missing elements within triplets to enrich the knowledge graph. Depending on the missing component, the task can be categorized into three subtypes: head entity completion $(?, r, t)$, tail entity completion $(h, r, ?)$, and relation completion $(h, ?, t)$. In this paper, we primarily focus on

---

[2]The $x$-axis denotes normalized score gap ($\frac{|\text{Score}_{\text{correct}} - \text{Score}_{\text{incorrect}}|}{std(\text{Score})}$) and $y$-axis indicates entity count per interval.

tail entity completion, following the setting of recent KG works [19, 20], as the other tasks can be reformulated into this one (See Appendix D.2).

**Related Works.** KG reasoning methods can be classified based on their use of structural information: message passing-based methods, which primarily leverage local structures, and transformer-based methods, which mainly exploit global structures. Message passing-based methods, such as [5, 27, 28], suffer from well-known limitations of message-passing networks, including incompleteness [29] and over-squashing [30]. Transformer-based methods, such as [31, 32], also have drawbacks. For example, they typically transform graph structures into sequential representations during knowledge encoding, potentially losing critical structural information inherent to KGs [33, 34]. Hybrid approaches that combine message-passing and transformers, such as [19, 20], leverage the strengths of both paradigms. However, they still face key challenges in effectively integrating and balancing local features learned via message passing with global KG information captured by self-attention. Besides, there are also triplet-based methods, such as TransE [35], ComplEx [36], DistMult [37], and RotatE [38], which treat triples as independent instances and often ignore graphs' topological structure. Beyond these categories, other approaches include meta-learning methods like MetaSD [31], rule pathbased models like RNNLogic [39], and tensor decomposition methods such as TuckER-IVR [40]. However, the optimization process of these methods does not directly take into account the issue of score over-smoothing. In response, we propose DuetGraph, explicitly addressing the challenge of score over-smoothing in KG reasoning.

# 3 DuetGraph

This section introduces DuetGraph, as shown in Figure 2[3]. The core architecture of DuetGraph consists of two components: a dual-pathway model for training (Steps ①-④), and a coarse-to-fine reasoning optimization for inference (Steps ⑤-⑧). Section 3.1 presents the dual-pathway global-local fusion model, and Section 3.2 details the coarse-to-fine reasoning optimization.

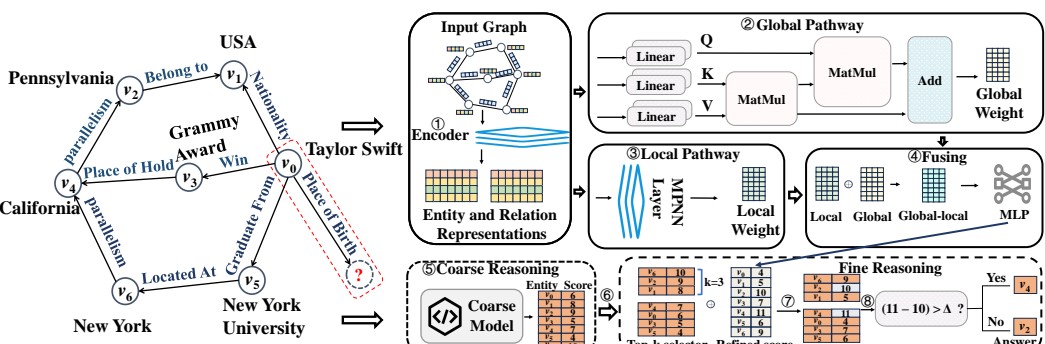

Figure 2: **Overview of DuetGraph**: ① Input KG to GNN encoder (e.g., GCN [15]) and output entity and relation representations; ② Employ a simple global attention mechanism [43] to compute the global weight; ③ Use the query-aware message passing networks [19] to compute the local weight; ④ Fuse the local and global weight using a multi-layer perceptron (MLP); ⑤ Use the coarse model (e.g., HousE [21] and RED-GNN [5]) to get the initial entity-to-score table; ⑥ Split the entity-to-score table into two subtables (i.e., high-score subtable and low-score subtable) based on Top-$k$ selector; ⑦ Update the two subtables based on the refined entity-to-score table predicted by dual-pathway global-local fusion model; ⑧ Output the answer based on the relationship between the maximum score gap of the two subsets and a predefined threshold $\Delta$.

---

[3]In dual-pathway model, after obtaining the representations in Step ④, we employ an MLP to transform each representation into a score. Loss function is defined for each training triplet $(h, r, t)$: $\mathcal{L} = -log(\sigma(t|h, r)) - \sum_{t'} log(1 - \sigma(t'|h, r))$, where $\sigma(\cdot|h, r)$ denotes the score of a candidate triplet, $t'$ denotes negative samples. Negative samples are generated by masking the correct answer and uniformly sampling with replacement from the remaining unmasked entities [41]. Finally, we update model parameters by optimizing negative sampling loss [38] with the Adam optimizer [42].

## 3.1 Dual-Pathway Global-Local Fusion Model

As previously mentioned, a single-pathway design is more likely to cause score over-smoothing, thereby impairing KG reasoning quality. Therefore, we decouple the message-passing networks and the transformer-based mechanism into two separate pathways, i.e., local pathway (Step ③) and global pathway (Step ②). Then, we fuse their outputs through an adaptive fusion model (Step ④). We detail the dual-pathway fusion model in the following paragraphs.

**Adaptive Global-Local Fusion.** A straightforward approach to achieve global-local information fusion is to simply sum the local and global weights. However, this method may fail to fully capture the complex interactions between local and global features, potentially hurting the model performance [44]. To address it, we introduce a learnable parameter $\alpha$ to adaptively assign weights to local and global information, enabling a more effective weighted fusion of the two components. Therefore, the final entity representation matrix $Z$ is computed as :

$$Z = \alpha \cdot Z_{\text{local}} + (1 - \alpha) \cdot Z_{\text{global}}, \tag{1}$$

where $Z_{\text{local}}$ denotes the local weight matrix, obtained through local pathway (Step ②), and $Z_{\text{global}}$ denotes the global weight matrix, obtained through global pathway (Step ③).

Then, the representation matrix $Z$ can be used for predicting the entity scores by an MLP (Step ④) .

**Theoretical Analysis.** Here, we theoretically show that our proposed dual-pathway fusion model offers superior alleviation of score over-smoothing compared to the single-pathway approach. To begin with, we give the upper bounds on entity score gap for both single-pathway and dual-pathway models in Lemma 1.

**Lemma 1** (**Upper Bounds on Score Gap for Different Models**)**.** *Let $\mathcal{M}_{\text{O}}$ denotes the weight matrix [45] of single-pathway model stacked with message passing and transformer, $\mathcal{M}_{\text{D}}$ denotes the weight matrix of our dual-pathway model. For any two entities $u, v \in \mathcal{V}$, the gap in their scores after $\ell$ layers of iteration can be bounded by:*

$$|S_u - S_v| \leq 2L_f(\sigma_{\max}(\mathcal{M}))^\ell \|\mathbf{X}^{(0)}\|_2, \qquad \mathcal{M} \in \{\mathcal{M}_{\text{O}}, \mathcal{M}_{\text{D}}\}, \tag{2}$$

*where $\sigma_{\max}(\mathcal{M})$ denotes the largest singular value of $\mathcal{M}$, $\mathbf{X}^{(0)}$ denotes initial entity feature matrix, $L_f$ is the Lipschitz constant [46], and $\|\cdot\|_2$ is Euclidean norm operation.*

We provide a detailed proof of Lemma 1 in Appendix A.1. Based on Theorem 1, we theoretically establish the relationship between the score gap and the weight matrix for each respective model. Specifically, we can get that the upper bound on score gap is related to the largest singular value of the weight matrix $\mathcal{M}$. To further scale the inequality in Equation 2, we derive the largest singular value upper bound of the weight matrix in Lemma 2.

**Lemma 2** (**Upper Bounds on Largest Singular Value**)**.** *For a weight matrix $\mathcal{M} \in \{\mathcal{M}_{\text{O}}, \mathcal{M}_{\text{D}}\}$, its largest singular value satisfies*

$$\sigma_{\max}(\mathcal{M}) < 1. \tag{3}$$

We provide a detailed proof of Lemma 2 in Appendix A.1. Based on Lemmas 1 and 2, as the number $\ell$ of iteration layers increases, the upper bound on score gaps **decrease exponentially** with respect to $\sigma_{\max}(\mathcal{M})$ because of the exponential functions properties. A larger $\sigma_{\max}(\mathcal{M})$ results in a greater upper bound and a slower decrease of it, suggesting that the model is more resistant to over-smoothing. Based on this, we further give the quantitative relationship between $\sigma_{\max}(\mathcal{M}_{\text{O}})$ and $\sigma_{\max}(\mathcal{M}_{\text{D}})$ in Lemma 3.

**Lemma 3** (**Relationship between $\sigma_{\max}(\mathcal{M}_{\text{O}})$ and $\sigma_{\max}(\mathcal{M}_{\text{D}})$** )**.** *Give the the learnable parameter $\alpha$ in Equation 1, the relationship between $\sigma_{\max}(\mathcal{M}_{\text{O}})$ and $\sigma_{\max}(\mathcal{M}_{\text{D}})$ is:*

$$\sigma_{\max}(\mathcal{M}_{\text{D}}) > \alpha - (1 - \alpha)\sigma_{\max}(\mathcal{M}_{\text{O}}). \tag{4}$$

We provide a detailed proof of Lemma 3 in Appendix A.1. Based on Lemmas 1, 2, and 3, we can derive the relationship between the learnable parameter $\alpha$ and the upper bound of the score gap, as shown in Theorem 1.

**Theorem 1** (**Relationship between $\alpha$ and Score Gap**)**.** *If $\alpha < \frac{\sigma_{\max}(\mathcal{M}_{\text{D}}) + \sigma_{\max}(\mathcal{M}_{\text{O}})}{1 + \sigma_{\max}(\mathcal{M}_{\text{O}})}$, the score gap upper bound of the the dual-pathway model is greater than that of the single-pathway model and the dual-pathway model shows a slower decrease of the score gap upper bound.*

We provide a detailed proof of Theorem 1 in Appendix A.1. Our adaptive fusion approach drives $\alpha$ below the theoretical threshold in Theorem 1 via parameter update with gradient descent. According to Theorem 1, the dual-pathway model outperforms the single-pathway model in mitigating over-smoothing.

**Complexity Analysis.** In this paragraph, we compare the time complexity of our dual-pathway fusion model with single-pathway approach. For a fair comparison, we assume both models have the same number of layers, including $L_m$ message passing layers and $L_t$ transformer layers. Under this setting, the overall time complexity of our dual-pathway fusion model and single-pathway approach is $\mathcal{O}(\max(L_m(|\mathcal{E}|\,d + |\mathcal{V}|\,d^2), L_t\,|\mathcal{V}|\,d^2))$ and $\mathcal{O}(L_m\,|\mathcal{E}|\,d + (L_m + L_t)\,|\mathcal{V}|\,d^2)$ respectively (we provide details in Appendix B.1.). Here, $|\mathcal{V}|$ and $|\mathcal{E}|$ respectively denote the number of entities and triplets and $d$ is the dimension of entity representation. In single-pathway approach, the message passing and transformer run sequentially, so their time complexity add together. By contrast, our dual-pathway fusion model processes them in parallel, so the overall complexity is only determined by the more expensive pathway. As a result, the dual-pathway model yields better time efficiency.

## 3.2 Coarse-to-Fine Reasoning Optimization

To address the over-smoothing issue caused by the one-stage approach as discussed in Section 1, we decompose the KG reasoning into two sequential stages: coarse stage and fine stage, as shown in Steps ⑤, ⑥, ⑦, and ⑧. We detail the design and implementation of each stage in the following paragraphs.

**Stage 1: Coarse-grained Reasoning.** In this stage, we first obtain an entity-to-score table by using the coarse model (Step ⑤). Then, the table is split into two subtables based on their rankings (Step ⑥): a high-score subtable made up of the topk entities, and a low-score subtable containing the remaining ones. The formal description of this process is as follows. Given a query $(h, r, ?)$, let $\mathcal{T} = \{(v, s_v) \mid v \in \mathcal{V}, s_v\}$ denote the full entity-to-score table, where $s_v$ is the score of entity $v$. $\mathrm{Rank}(v)$ denotes the rank of entity $v$ in descending order of the scores. Accordingly, we split $\mathcal{T}$ into two subtables as follows:

$$\mathcal{T}^{\mathrm{high}} = \{(v, s_v) \in \mathcal{T} : \mathrm{Rank}(v) \leq k\}, \qquad \mathcal{T}^{\mathrm{low}} = \{(v, s_v) \in \mathcal{T} : \mathrm{Rank}(v) > k\},$$

where $k$ is a hyperparameter controlling the cutoff rank. $\mathcal{T}^{\mathrm{high}}$ is the high-score subtable and $\mathcal{T}^{\mathrm{low}}$ is the low-score subtable.

**Stage 2: Fine-grained Reasoning.** At this stage, we firstly update $\mathcal{T}^{\mathrm{high}}$ and $\mathcal{T}^{\mathrm{low}}$ with the fine model (i.e., the dual-pathway model introduced in Section 3). Then, we extract the entities with the highest score from each subtables, denoted as $(e_h, s_{e_h})$ for $\mathcal{T}^{\mathrm{high}}$ and $(e_l, s_{e_l})$ for $\mathcal{T}^{\mathrm{low}}$. After that, we compute the difference $\gamma = s_{e_l} - s_{e_h}$ based on the pre-defined threshold $\Delta$. If $\gamma$ exceeds $\Delta$, this indicates that the highest-score entity in the low-score subtable clearly surpasses the highest-score entity in the high-score subtable. Therefore, we select the highest-score entity from $\mathcal{T}^{\mathrm{low}}$ as the final answer; otherwise, we select that from $\mathcal{T}^{\mathrm{high}}$.

By introducing this adjustable threshold $\Delta$, we enable entities from both the high-score and low-score subtables to be dynamically selected as the answer. This design enhances flexibility and reduces selection bias in the decision process.

**Theoretical Analysis.** In coarse-to-fine optimization, since the final prediction is made based on comparing the highest scores from the high-score and low-score subtables. Thus, the score gap is particularly crucial for mitigating over-smoothing and we theoretically demonstrate how the coarse-to-fine optimization mitigates over-smoothing by amplifying the gap between the highest scores in the two subtables, as shown in Theorem 2.

**Theorem 2** (**Lower Bound on Score Gap Between High-score and Low-score Subtables**). *The lower bound on the expected gap between the top scores of the two subtables ($s_{e_h}$ and $s_{e_l}$) is:*

$$\mathbb{E}[|s_{e_h} - s_{e_l}|] > \left| \left( \frac{1}{N_h^2 + 1} - \frac{1}{(N_l^2 + 1)} \right) \cdot \sigma \right|, \tag{5}$$

*where $N_h$ and $N_l$ denote the number of entities in high-score and low-score subtables respectively. $\sigma$ is the standard deviation of the entity score.*

We provide a detailed proof of Theorem 2 in Appendix A.2. Based on Theorem 2, we establish the relationship between the score gap and the number of entities. In our setup, the ratio of the number of entities in the low-score subtable to those in the high-score subtable exceeds 1,000. Therefore, based on Theorem 2, we can derive that the lower bound on the expected gap between the top scores of the two subtables is more than $0.1\sigma$ (Detailed proof of this in Appendix A.2). In comparision, other baseline methods (as shown in Figure 1) exhibit score gaps between correct and incorrect answers are typically less than $0.02\sigma$. This demonstrates that our optimization can amplifying the score gap, thus mitigating over-smoothing. Building on this, we additionally present Theorem 3 to theoretically demonstrate that coarse-to-fine optimization also improves the quality of KG reasoning.

**Theorem 3** (**Effectiveness of Coarse-to-Fine Optimization**). *Let $P$ and $P'$ denotes the probabilities of correctly identifying the answer with and without coarse-to-fine optimization, respectively. Then, we have $P > P'$.*

We provide a detailed proof of Theorem 3 in Appendix A.3.

**Complexity Analysis.** In this paragraph, we compare the complexity of our coarse-to-fine stage with one-stage approach. The time complexities of coarse-to-fine stage and one-stage are $\mathcal{O}(\max(L_m(|\mathcal{E}|\,d + |\mathcal{V}|\,d^2), L_t\,|\mathcal{V}|\,d^2) + |\mathcal{V}|\,log\,|\mathcal{V}|)$ and $\mathcal{O}(\max(L_m(|\mathcal{E}|\,d + |\mathcal{V}|\,d^2), L_t\,|\mathcal{V}|\,d^2))$ , respectively. We provide detail proof of these in Appendix B.2. Here, $|\mathcal{V}|$ and $|\mathcal{E}|$ denote the number of entities and triplets, respectively. $L_m$ and $L_t$ represent the number of message passing layers and transformer layers. $d$ is the dimension of the entity representation. In practice, $|\mathcal{V}|\,log\,|\mathcal{V}|$ is much smaller than $|\mathcal{V}|\,d^2$. For example, in FB15k-237 dataset [47], the number of entities is 14,541, and the representation dimension is 32. Accordingly, $|\mathcal{V}|\,log\,|\mathcal{V}|$ is approximately $10^5$ and $|\mathcal{V}|\,d^2$ is approximately $10^7$. Therefore, the time complexity of the coarse-to-fine stage remains comparable to that of the one-stage.

# 4   Empirical Evaluation

In this section, we conduct extensive experiments to answer the following research questions: **(RQ1)** Can DuetGraph effectively improve the performance of inductive KG reasoning tasks? **(RQ2)** Can DuetGraph effectively improve the performance of transductive KG reasoning tasks? **(RQ3)** Can DuetGraph demonstrate strong scalability in KG reasoning tasks by achieving high training efficiency? **(RQ4)** How is the effectiveness of the components of DuetGraph? **(RQ5)** In the coarse-to-fine reasoning, what is the standard of the coarse model? **(RQ6)** Is DuetGraph sensitive to hyperparameter $k$, where $k$ denotes the number of entities in a high-score subset? **(RQ7)** How generalizable is DuetGraph across tasks on knowledge graphs?

## 4.1   Experiments setup

**Inductive Datasets.** For inductive reasoning, following Liu et al. [19], we use the same data divisions of FB15k-237 [47], WN18RR [48], and NELL-995 [49]. Each division consists of 4 versions, resulting in 12 subsets in total. Notably, in each subset, the training and test sets contain disjoint sets of entities while sharing the same set of relations.

**Transductive Datasets.** For transductive reasoning, we conduct experiments on four widely utilized KG reasoning datasets: FB15k-237  [47], WN18RR [48], NELL-995 [49], and YAGO3-10 [50], adopting the standard data splits provided by prior works [28, 51].

**Triple Classification Datasets.** For the triple classification task, we conduct experiments on three widely used knowledge graph datasets: UMLS[52], FB13[53] and WN11[53].

**Triple Classification Baselines.** The following four categories of SOTA models are adopted as baselines for comparison with DuetGraph in triple classification: *triplet-based (HousE [21])*, *message passing-based (AdaProp [51])*, *transformer-based (HittER [54])*, and *hybrid message passing-transformer (KnowFormer [19])* models (SOTA methods for comprehensive comparison).

Table 1: Inductive KG reasoning performance for various methods on 12 subsets. (The best results are bolded in red with a yellow highlight. Second-best results are with a blue highlight. Results are either sourced directly from original papers or reproduced based on available code.)

| Method | v1 | | | v2 | | | v3 | | | v4 | | |
|---|---|---|---|---|---|---|---|---|---|---|---|---|
| | MRR | H@1 | H@10 | MRR | H@1 | H@10 | MRR | H@1 | H@10 | MRR | H@1 | H@10 |
| **FB15k-237** | | | | | | | | | | | | |
| DRUM [55] | 0.333 | 24.7 | 47.4 | 0.395 | 28.4 | 59.5 | 0.402 | 30.8 | 57.1 | 0.410 | 30.9 | 59.3 |
| NBFNet [28] | 0.442 | 33.5 | 57.4 | 0.514 | 42.1 | 68.5 | 0.476 | 38.4 | 63.7 | 0.453 | 36.0 | 62.7 |
| RED-GNN [5] | 0.369 | 30.2 | 48.3 | 0.469 | 38.1 | 62.9 | 0.445 | 35.1 | 50.3 | 0.442 | 34.0 | 62.1 |
| A*Net [13] | 0.457 | 38.1 | 58.9 | 0.510 | 41.9 | 67.2 | 0.476 | 38.9 | 62.9 | 0.466 | 36.5 | 64.5 |
| AdaProp [51] | 0.310 | 19.1 | 55.1 | 0.471 | 37.2 | 65.9 | 0.471 | 37.7 | 63.7 | 0.454 | 35.3 | 63.8 |
| Ingram [56] | 0.293 | 16.7 | 49.3 | 0.274 | 16.3 | 48.2 | 0.233 | 14.0 | 40.8 | 0.214 | 11.4 | 39.7 |
| KnowFormer [19] | 0.466 | 37.8 | 60.6 | 0.532 | 43.3 | 70.3 | 0.494 | 40.0 | 65.9 | 0.480 | 38.3 | 65.3 |
| DuetGraph (Ours) | **0.507** | **42.7** | **63.2** | **0.549** | **44.8** | **72.9** | **0.518** | **42.3** | **69.9** | **0.501** | **39.8** | **67.0** |
| **WN18RR** | | | | | | | | | | | | |
| DRUM [55] | 0.666 | 61.3 | 77.7 | 0.646 | 59.5 | 74.7 | 0.380 | 33.0 | 47.7 | 0.627 | 58.6 | 70.2 |
| NBFNet [28] | 0.741 | 69.5 | 82.6 | 0.704 | 65.1 | 79.8 | 0.452 | 39.2 | 56.8 | 0.641 | 60.8 | 69.4 |
| RED-GNN [5] | 0.701 | 65.3 | 79.9 | 0.690 | 63.3 | 78.0 | 0.427 | 36.8 | 52.4 | 0.651 | 60.6 | 72.1 |
| A*Net [13] | 0.727 | 68.2 | 81.0 | 0.704 | 64.9 | 80.3 | 0.441 | 38.6 | 54.4 | 0.661 | 61.6 | 74.3 |
| AdaProp [51] | 0.733 | 66.8 | 80.6 | 0.715 | 64.2 | 82.6 | 0.474 | 39.6 | 58.8 | 0.662 | 61.1 | 75.5 |
| Ingram [56] | 0.277 | 13.0 | 60.6 | 0.236 | 11.2 | 48.0 | 0.230 | 11.6 | 46.6 | 0.118 | 4.1 | 25.9 |
| SimKGC [57] | 0.315 | 19.2 | 56.7 | 0.378 | 23.9 | 65.0 | 0.303 | 18.6 | 54.3 | 0.308 | 17.5 | 57.7 |
| KnowFormer [19] | 0.752 | 71.5 | 81.9 | 0.709 | 65.6 | 81.7 | 0.467 | 40.6 | 57.1 | 0.646 | 60.9 | 72.7 |
| DuetGraph (Ours) | **0.758** | **72.1** | 81.7 | **0.719** | **66.7** | 81.1 | **0.501** | **44.3** | **62.2** | **0.662** | **62.1** | 73.1 |
| **NELL-995** | | | | | | | | | | | | |
| NBFNet [28] | 0.584 | 50.0 | 79.5 | 0.410 | 27.1 | 63.5 | 0.425 | 26.2 | 60.6 | 0.287 | 25.3 | 59.1 |
| RED-GNN [5] | 0.637 | 52.2 | 86.6 | 0.419 | 31.9 | 60.1 | 0.436 | 34.5 | 59.4 | 0.363 | 25.9 | 60.7 |
| AdaProp [51] | 0.644 | 52.2 | 88.6 | 0.452 | 34.4 | 65.2 | 0.435 | 33.7 | 61.8 | 0.366 | 24.7 | 60.7 |
| Ingram [56] | 0.697 | 57.5 | 86.5 | 0.358 | 25.3 | 59.6 | 0.308 | 19.9 | 50.9 | 0.221 | 12.4 | 44.0 |
| KnowFormer [19] | 0.827 | 77.0 | 93.0 | 0.465 | 35.7 | 65.7 | 0.478 | 37.8 | 65.7 | 0.378 | 26.7 | 59.8 |
| DuetGraph (Ours) | **0.850** | **78.5** | **96.5** | **0.543** | **44.4** | **69.1** | **0.535** | **43.2** | **72.6** | **0.464** | **35.4** | **68.4** |

**Inductive Baselines.** We compare DuetGraph with 8 baseline methods for inductive KG reasoning as shown in Table 1. For completeness, we note that some baselines do not support certain datasets due to limitations in their released code. Details are provided in Appendix C.2.

**Transductive Baselines.** The following four categories of SOTA models are adopted as baselines for comparison with DuetGraph in transductive KG reasoning: *triplet-based models*, *message passing-based models*, *transformer-based models*, *hybrid message passing-transformer models (including our proposed method and KnowFormer [19])* and other approaches, as shown in Table 2.

**Evaluations Metrics.** The model performance is measured by **Mean Reciprocal Rank (MRR)** [35] and **Hit Rate at** $k$ (Hits@$k$, where $k \in \{1, 10\}$) [35]. Hits@$k$ assesses whether the true entity of a triplet appears within the top-$k$ ranked candidate entities. If the true entity is ranked $k$ or higher, the result is recorded as 1: otherwise, it is recorded as 0. Metrics are formalized as follows.

Hits@$k = \frac{1}{|\mathcal{T}_{\text{test}}|} \sum_{t_i \in \mathcal{T}_{\text{test}}} f(\text{rank}_i)$, where $f(x) = \begin{cases} 1, & x \leq k \\ 0, & x > k \end{cases}$ and $\mathcal{T}_{\text{test}}$ is the test set containing $|\mathcal{T}_{\text{test}}|$ triplets. Each $t_i$ is the $i$-th test triplet, and $\text{rank}_i$ represents the position of the correct entity in the ranked list of candidates. MRR is calculated as the average of the reciprocals of the ranks assigned to the correct entities in the prediction results. MRR $= \frac{1}{|\mathcal{T}_{\text{test}}|} \sum_{t_i \in \mathcal{T}_{\text{test}}} \left( \frac{1}{\text{rank}_i} \right)$, where $\mathcal{T}_{\text{test}}$ is the test set, and $\text{rank}_i$ represents the rank of the true entity in the candidate list for $t_i$.

## 4.2 Performance

To answer (**RQ1**), we evaluate DuetGraph on 12 datasets. The results, shown in Table 1, demonstrate the strong performance of DuetGraph compared to baseline models. Specifically, DuetGraph surpass SOTA methods by up to 8.6% improvement in MRR, 8.7% improvement in Hits@1, and 7.7% improvement in Hits@10. It ranks first in Hits@1 on every evaluated version (v1–v4) of the FB15k-

Table 2: Transductive KG reasoning performance for various methods on 4 datasets. (The best results are bolded in red with a yellow highlight. Second-best results are with a blue highlight. Results are either sourced directly from original papers or reproduced based on publicly available code. "-" indicates unavailable results due to insufficient information for reproduction. )

| Method | FB15k-237 | | | WN18RR | | | NELL-995 | | | YAGO3-10 | | |
|---|---|---|---|---|---|---|---|---|---|---|---|---|
| | MRR | H@1 | H@10 | MRR | H@1 | H@10 | MRR | H@1 | H@10 | MRR | H@1 | H@10 |
| **Triplet-based** | | | | | | | | | | | | |
| TransE [35] | 0.330 | 23.2 | 52.6 | 0.222 | 1.4 | 52.8 | 0.507 | 42.4 | 64.8 | 0.510 | 41.3 | 68.1 |
| DistMult [37] | 0.358 | 26.4 | 55.0 | 0.455 | 41.0 | 54.4 | 0.510 | 43.8 | 63.6 | 0.566 | 49.1 | 70.4 |
| RotatE [38] | 0.337 | 24.1 | 53.0 | 0.477 | 42.8 | 57.1 | 0.508 | 44.8 | 60.8 | 0.495 | 40.2 | 67.0 |
| HousE [21] | 0.361 | 26.6 | 55.1 | 0.511 | 46.5 | 60.2 | 0.519 | 45.8 | 61.8 | 0.571 | 49.1 | 71.4 |
| **Message passing-based** | | | | | | | | | | | | |
| CompGCN [14] | 0.355 | 26.4 | 53.5 | 0.479 | 44.3 | 54.6 | 0.463 | 38.3 | 59.6 | 0.421 | 39.2 | 57.7 |
| NBFNet [28] | 0.415 | 32.1 | 59.9 | 0.551 | 49.7 | 66.6 | 0.525 | 45.1 | 63.9 | 0.563 | 48.0 | 70.8 |
| RED-GNN [5] | 0.374 | 28.3 | 55.8 | 0.533 | 48.5 | 62.4 | 0.543 | 47.6 | 65.1 | 0.556 | 48.3 | 68.9 |
| A*Net [13] | 0.411 | 32.1 | 58.6 | 0.549 | 49.5 | 65.9 | 0.521 | 44.7 | 63.1 | 0.556 | 47.0 | 70.7 |
| AdaProp [51] | 0.417 | 33.1 | 58.5 | 0.562 | 49.9 | 67.1 | 0.554 | 49.3 | 65.5 | 0.573 | 51.0 | 68.5 |
| ULTRA [58] | 0.368 | 27.2 | 56.4 | 0.480 | 41.4 | 61.4 | 0.509 | 44.1 | 66.0 | 0.557 | 47.1 | 71.0 |
| **Transformer-based** | | | | | | | | | | | | |
| HittER [54] | 0.373 | 27.9 | 55.8 | 0.503 | 46.2 | 58.4 | 0.518 | 43.7 | 65.9 | 0.339 | 25.1 | 50.8 |
| KGT5 [59] | 0.276 | 21.0 | 41.4 | 0.508 | 48.7 | 54.4 | - | - | - | 0.426 | 36.8 | 52.8 |
| N-Former [60] | 0.373 | 27.9 | 55.6 | 0.489 | 44.6 | 58.1 | - | - | - | - | - | - |
| SAttLE [22] | 0.360 | 26.8 | 54.5 | 0.491 | 45.4 | 55.8 | 0.512 | 42.2 | 66.0 | 0.475 | 36.7 | 68.2 |
| **Others** | | | | | | | | | | | | |
| MetaSD [31] | 0.391 | 30.0 | 57.1 | 0.491 | 44.7 | 57.0 | 0.516 | 45.5 | 61.5 | OOM | OOM | OOM |
| RNNLogic [39] | 0.344 | 25.2 | 53.0 | 0.483 | 44.6 | 55.8 | 0.516 | 46.3 | 57.8 | 0.554 | 50.9 | 62.2 |
| TuckeER-IVR [40] | 0.368 | 27.4 | 55.5 | 0.501 | 46.0 | 57.9 | 0.505 | 42.8 | 63.7 | 0.581 | 50.8 | 71.2 |
| **Hybrid** | | | | | | | | | | | | |
| KnowFormer [19] | 0.430 | 34.3 | 60.8 | 0.579 | 52.8 | 68.7 | 0.566 | 50.2 | 67.5 | 0.615 | 54.7 | 73.4 |
| DuetGraph (Ours) | **0.453** | **36.1** | **62.4** | **0.593** | **54.2** | **69.9** | **0.590** | **52.1** | **71.2** | **0.631** | **56.1** | **74.8** |

237, WN18RR, and NELL-995 datasets, indicating its strong ability to accurately predict the correct entity at the top rank. Importantly, our method relies solely on the structural information of the KG, without relying on external textual features, highlighting its strong generalization capability.

To answer (**RQ2**), we evaluate the performance of DuetGraph on four widely utilized transductive KG reasoning datasets. Table 2 demonstrates the impressive performance of DuetGraph compared to baseline models. Specifically, DuetGraph demonstrates substantial performance gains over baseline methods, with improvements of up to 37.1% in MRR, 52.8% in Hits@1, and 24.0% in Hits@10.

It is worth noting that in inductive KG reasoning, the entities to be predicted are unseen during training, which not only aligns more closely with real-world scenarios but also poses a greater challenge. Based on these results, DuetGraph exhibits remarkable generalization and adaptability.

## 4.3 Efficiency

To answer (**RQ3**), we evaluate Hits@1 throughout training on the FB15k-237 and YAGO3-10, the latter being a large-scale dataset with millions of training triples. We compare DuetGraph with the best models in each category—AdaProp (message passing-based), SAttLE (transformer-based), and KnowFormer (hybrid). As shown in Figure 3, DuetGraph finally achieves SOTA performance on FB15k-237 while reducing training time by nearly 50% compared to the second-best method. We also observe from Figure 3 that DuetGraph achieves SOTA performance on YAGO3-10 while requiring less training time compared to other methods. These results show that DuetGraph achieves scalability through high training efficiency. The observed improvement is primarily attributable to this dual-pathway design, which enables parallel training of both local and global pathways.

Furthermore, to demonstrate the efficiency of DuetGraph on very large KGs, we conduct experiments on Wikidata5M [61] and Freebase [62], and compare DuetGraph with highly efficient rule-based methods (e.g., AnyBURL [63]). It is worth noting that for the Wikidata5M and Freebase datasets, the AnyBURL paper [63] does not explicitly specify under which data split the results in Table 2 of [63] were obtained. For a fair comparison, we use the same data split as in [62].

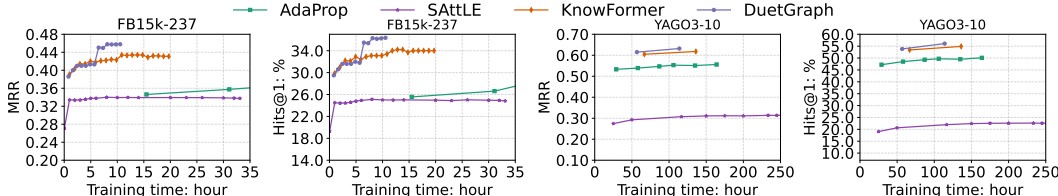

Figure 3: Hits@1 and MRR w.r.t. time on FB15k-237 and YAGO3-10.

Table 3: Comparison of different methods across very large knowledge graphs. (The best results are bolded in red with a yellow highlight.)

| Method | Wikidata5M | | | | | Freebase | | | | |
|---|---|---|---|---|---|---|---|---|---|---|
| | MRR | H@1 | H@10 | Learning | Inference | MRR | H@1 | H@10 | Learning | Inference |
| AnyBURL (Rule-based) | 0.350 | 30.9 | 42.9 | 10,000s | 4,462s | 0.588 | 53.6 | 68.2 | 10,000s | 3,142s |
| KnowFormer (Emb-based) | 0.332 | 26.7 | 46.3 | 31,436s | 105s | 0.684 | 65.7 | 73.6 | 32,109s | 176s |
| DuetGraph (Emb-based) | **0.363** | **32.7** | **49.5** | 28,866s | **80s** | **0.697** | **69.3** | **73.8** | 30,158s | **141s** |

As shown in Table 3, DuetGraph achieves SOTA quality performance with a learning time in the same order of magnitude as AnyBURL [63], and it demonstrates a significant reduction in inference time compared to AnyBURL [63]. It demonstrates that DuetGraph maintains high efficiency and strong quality even when applied to extremely large knowledge graphs.

## 4.4 Ablation Study

To address (**RQ4**), we perform an ablation study by removing key components of DuetGraph: (1) the local pathway, (2) the global pathway, (3) the coarse-to-fine reasoning optimization (i.e., reducing DuetGraph model to the dual-pathway fusion alone), (4) the dual-pathway fusion model (i.e., leaving only the coarse-grained reasoning), and (5) the threshold $\Delta$ in the fine-grained stage, which prevents correction for low-score predictions.

As shown in Table 4, removing either the local or global pathway degrades performance, confirming the necessity of both information types. Eliminating coarse-to-fine reasoning leads to a notable drop in Hits@1, demonstrating its effectiveness in refining predictions. Excluding the dual-pathway module results in the largest performance loss, which underscores its crucial role in reasoning performance. Finally, removing the threshold $\Delta$ reduces accuracy due to uncorrected errors in cases where the correct entity is excluded from the high-score subset during coarse reasoning.

To answer (**RQ5**), we additionally evaluate the performance of DuetGraph on four transductive KG reasoning datasets using three different types of coarse-grained reasoning models: a triplet-based model (HousE [21]), a message passing-based model (RED-GNN [5]), and a hybrid message passing-transformer model (KnowFormer [19]). The triplet-based model focuses exclusively on local triple-level patterns. The message passing model captures neighborhood-level information, and the transformer model handles global patterns (like our fine model).

As shown in Table 5, DuetGraph, when integrated with any of the three coarse-grained models, consistently outperforms its competitors. Among them, the triplet-based model achieves the best performance as a coarse model. This aligns with prior work [64], which shows that maximizing architectural diversity between coarse and fine models leads to better overall performance; in this case, the triplet-based model benefits from its maximal architectural difference from our global-information-focused fine model.

## 4.5 Parameter Analysis

To answer (**RQ6**), we conduct experiments by varying the hyperparameter $k$ introduced in Section 3.2 across four different transductive datasets. As shown in Figure 4, DuetGraph is insensitive to the parameter $k$, suggesting stable performance. We further conduct hyperparameter experiments on inductive datasets, as presented in Appendix D.4, and obtain consistent results.

Table 4: Different components ablation study of DuetGraph on 4 transductive KG reasoning datasets. (The best results are bolded in red with a yellow highlight.)

| Method | FB15k-237 | | | WN18RR | | | NELL-995 | | | YAGO3-10 | | |
|---|---|---|---|---|---|---|---|---|---|---|---|---|
| | MRR | H@1 | H@10 | MRR | H@1 | H@10 | MRR | H@1 | H@10 | MRR | H@1 | H@10 |
| **DuetGraph** | **0.453** | **36.1** | **62.4** | **0.593** | **54.2** | **69.9** | **0.590** | **52.1** | **71.2** | **0.631** | **56.1** | **74.8** |
| *w/o* local | 0.445 | 35.1 | 61.2 | 0.584 | 54.1 | 69.0 | 0.582 | 51.0 | 70.3 | 0.617 | 54.2 | 73.7 |
| *w/o* global | 0.441 | 34.9 | 61.4 | 0.565 | 51.7 | 66.6 | 0.586 | 51.0 | 69.8 | 0.614 | 53.8 | 74.4 |
| *w/o* Coarse-to-Fine reasoning | 0.437 | 34.8 | 61.1 | 0.580 | 53.0 | 68.9 | 0.567 | 50.5 | 67.7 | 0.616 | 54.8 | 73.5 |
| *w/o* Dual-Pathway fusion model | 0.355 | 25.9 | 54.7 | 0.512 | 46.6 | 60.6 | 0.534 | 46.6 | 51.2 | 0.563 | 48.4 | 70.7 |
| *w/o* threshold value $\Delta$ | 0.395 | 31.7 | 55.8 | 0.551 | 48.9 | 66.1 | 0.544 | 49.8 | 66.5 | 0.595 | 53.4 | 70.9 |

Table 5: Different coarse-grained model ablation study of DuetGraph on 4 transductive KG reasoning datasets. (The best results are bolded in red with a yellow highlight.)

| Method | FB15k-237 | | | WN18RR | | | NELL-995 | | | YAGO3-10 | | |
|---|---|---|---|---|---|---|---|---|---|---|---|---|
| | MRR | H@1 | H@10 | MRR | H@1 | H@10 | MRR | H@1 | H@10 | MRR | H@1 | H@10 |
| DuetGraph (w/ triplets-based model as coarse model) | **0.453** | **36.1** | **62.4** | **0.593** | **54.2** | **69.9** | **0.590** | **52.1** | **71.2** | **0.631** | **56.1** | **74.8** |
| DuetGraph (w/ message passing-based model as coarse model) | 0.446 | 35.4 | 61.3 | 0.589 | 53.5 | 69.0 | 0.584 | 51.7 | 70.2 | 0.622 | 55.4 | 73.9 |
| DuetGraph (w/ transformer-based model as coarse model) | 0.445 | 34.8 | 62.4 | 0.586 | 53.2 | 69.4 | 0.579 | 50.8 | 70.5 | 0.624 | 55.7 | 74.2 |

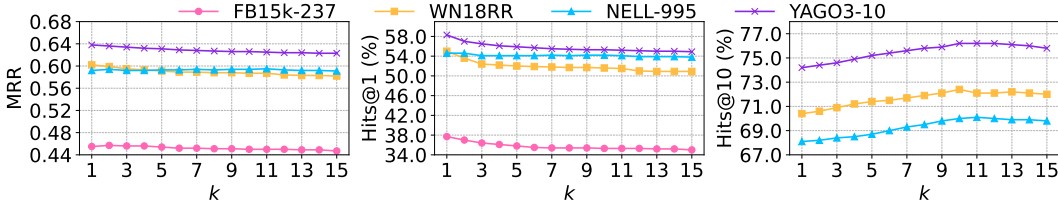

Figure 4: Effect of $k$ on the performance metrics of KG reasoning for different datasets (Transductive).

## 4.6 Generalization

To answer (**RQ7**), we evaluate DuetGraph's performance on the triple classification task. The experimental results in Table 6 demonstrate that DuetGraph consistently outperforms all baseline methods across all datasets, achieving new SOTA performance on the triple classification task, which further highlights the task generality of our proposed DuetGraph framework.

Table 6: Comparison of triple classification accuracy on different datasets. (The best results are bolded in red with a yellow highlight.)

| Method | UMLS Acc (%) | FB13 Acc (%) | WN11 Acc (%) |
|---|---|---|---|
| HousE [21] | 83.1 | 69.8 | 65.3 |
| HittER [54] | 59.4 | 62.2 | 69.6 |
| AdaProp [51] | 77.0 | 71.9 | 67.1 |
| KnowFormer [19] | 83.1 | 77.3 | 70.2 |
| DuetGraph | **84.3** | **80.0** | **71.9** |

## 5 Conclusion

This paper proposes DuetGraph, a coarse-to-fine KG reasoning mechanism with dual-pathway global-local fusion to alleviate score over-smoothing in KG reasoning. DuetGraph mitigates over-smoothing by allocating the processing of local (via message passing) and global (via attention) information to two distinct pathways, rather than stacking them. This design prevents mutual interference and preserves representational discrimination. Experimental results show that DuetGraph outpeforms SOTA baselines on both quality and training efficiency.

**Acknowledgements:** This work was supported by the National Natural Science Foundation of China (NSFC) under Grant 62472400.

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

# A  Proofs of Theorems

In this section, we provide the theorem proofs in method part, including 1) why our proposed dual-pathway global-local fusion model can alleviate the over-smoothing in KG; 2) why our proposed coarse-to-fine reasoning optimization can alleviate the over-smoothing in KG; 3) why our proposed coarse-to-fine reasoning optimization can improve the quality of KG reasoning.

## A.1  Dual-Pathway Global-Local Fusion Model Effectively Alleviating Over-Smoothing.

*Proof.* Let entity initial representation be denoted as $X^{(0)} \in \mathbb{R}^{n \times d}$, symmetrically normalized adjacency matrix in message passing networks is denoted as $A \in \mathbb{R}^{n \times n}$. The attention matrix computed by a single layer of global attention is denoted as $P \in \mathbb{R}^{n \times n}$. We construct a weight matrix of one-pathway model stacked with $L$ message passing layers and a transformer layer defined as:

$$\mathcal{M}_O = PA^L \tag{6}$$

We construct a weight matrix of dual-pathway fusion model defined as:

$$\mathcal{M}_D = \alpha A^L + (1-\alpha)P \tag{7}$$

where $\alpha$ is the learnable parameter in Equation 1. We first focus on the entity representation obtained after $\ell$ iterations:

$$X^{(\ell)} = \mathcal{M}^\ell X^{(0)}, \mathcal{M} \in \{\mathcal{M}_O, \mathcal{M}_D\} \tag{8}$$

According to basic properties of the matrix paradigm, we can get spectral norm of $X^{(m)}$ satisfies

$$\|X^{(\ell)}\|_2 = \|\mathcal{M}^\ell X^{(0)}\|_F \le \|\mathcal{M}^\ell\|_2 \|X^{(0)}\|_2 \le (\sigma_{\max}(\mathcal{M}))^\ell \|X^{(0)}\|_2 \tag{9}$$

The representation discrepancy between any two entities $u$ and $v$ satisfies

$$\|x_u^{(\ell)} - x_v^{(\ell)}\|_2 \le \|x_u^{(\ell)}\|_2 + \|x_v^{(\ell)}\|_2 \le 2\|X^{(\ell)}\|_2 \le 2 (\sigma_{\max}(\mathcal{M}))^\ell \|X^{(0)}\|_2 \tag{10}$$

Then, entity representations are mapped to scalar scores through a multilayer perceptron (MLP). According to the principle of Lipschitz continuity, the score gap between any two entities can be bounded by

$$|S_u - S_v| \le L_f \cdot \|x_u^{(\ell)} - x_v^{(\ell)}\|_2 \le 2L_f(\sigma_{\max}(\mathcal{M}))^\ell \|\mathbf{X}^{(0)}\|_2 \tag{11}$$

where $f : \mathbb{R}^d \to \mathbb{R}$ denotes the MLP and $L_f$ is the Lipschitz constant [46]. Since $A$ is a symmetrically normalized adjacency matrix, spectral theory ensures that its eigenvalues $\{\lambda_i\}_{i=1}^n$ satisfy $1 = \lambda_1 > \lambda_2 \ge \cdots \ge \lambda_n > -1$. Hence, we conclude that largest eigenvalue of $A^L$ is equal to 1, and its largest singular value $\sigma_{\max}(A^L)$ is equal to 1.

Since the attention matrix $P$ is row-stochastic, its largest eigenvalue is 1, and all other eigenvalues satisfy $|\mu_i| < 1$. According to the definition of singular values, it follows that the largest singular value $\sigma_{\max}(P)$ of $P$ is less than 1.

Therefore, we can get

$$\sigma_{\max}(\mathcal{M}_O) = \sigma_{\max}(PA^L) \le \sigma_{\max}(P) \cdot \sigma_{\max}(A^L) = \sigma_{\max}(P) < 1 \tag{12}$$

and since the spectral norm and the maximum singular value are equal we can get

$$\sigma_{\max}(\mathcal{M}_D) = \|\alpha A^L + (1-\alpha)P\|_2 \le \|\alpha A^L\|_2 + \|(1-\alpha)P\|_2 \le \alpha\|A^L\|_2 + (1-\alpha)\|P\|_2 < 1 \tag{13}$$

Then, we have

$$\sigma_{\max}(\mathcal{M}) < 1, \qquad \mathcal{M} \in \{\mathcal{M}_O, \mathcal{M}_D\} \tag{14}$$

Since the spectral norm and the maximum singular value are equal, we can use the inverse triangle inequality to derive the following:

$$\sigma_{\max}(\mathcal{M}_D) = \|\alpha A^L + (1-\alpha)P\|_2 \ge |\|\alpha A^L\|_2 - \|(1-\alpha)P\|_2| = |\alpha \cdot 1 - (1-\alpha)\sigma_{\max}(P)| \tag{15}$$

According to Equation 12, we have

$$\sigma_{\max}(\mathcal{M}_D) \ge \alpha - (1-\alpha)\sigma_{\max}(P) \ge \alpha - (1-\alpha)\sigma_{\max}(\mathcal{M}_O) \tag{16}$$

Therefore, as long as the learnable parameter $\alpha$ is less than $\frac{\sigma_{\max}(\mathcal{M}_D) + \sigma_{\max}(\mathcal{M}_O)}{1 + \sigma_{\max}(\mathcal{M}_O)}$, $\sigma_{\max}(\mathcal{M}_D)$ will necessarily be greater than $\sigma_{\max}(\mathcal{M}_O)$. The result of Equation 11 indicates that as the number of iterations $\ell$ increases, the score gap between any two entities will decrease exponentially with respect $\sigma_{\max}(\mathcal{M})$. This implies that If $\alpha < \frac{\sigma_{\max}(\mathcal{M}_D) + \sigma_{\max}(\mathcal{M}_O)}{1 + \sigma_{\max}(\mathcal{M}_O)}$, the score gap upper bound of the dual-pathway model is greater than that of the one-pathway model and the dual-pathway model shows a slower decrease in the score gap upper bound. Consequently, dual-pathway model effectively mitigates the oversmoothing. $\square$

## A.2 Coarse-to-Fine Reasoning Optimization Alleviates the over-smoothing in KG.

*Proof.* For a set of scores $\{s_1, s_2, ..., s_k\}$, set a score threshold $t = \mu' + \frac{\sigma'}{k}$, where $\mu'$ is the mean of this set of scores and $\sigma'$ is the standard deviation of this set of scores. Using Cantelli inequality,

$$P(s_i \geq t) = P(s_i \geq \mu' + \frac{\sigma'}{k}) \geq \frac{(\frac{\sigma'}{k})^2}{\sigma'^2 + (\frac{\sigma'}{k})^2} = \frac{1}{k^2 + 1} \tag{17}$$

The probability that at least one $s_i$ of the $k$ scores is more than $t$ is

$$P(\max_i s_i \geq t) \geq 1 - \left(1 - \frac{1}{k^2 + 1}\right)^k \approx \frac{k}{k^2 + 1} \tag{18}$$

Therefore,

$$\mathbb{E}[\max s_i] \geq t \cdot P(\max_i s_i \geq t) \geq (\mu' + \frac{\sigma'}{k}) \cdot P(\max_i s_i \geq t) = (\mu' + \frac{\sigma'}{k}) \cdot \frac{k}{k^2 + 1} \tag{19}$$

Let the scores of candidate entities in fine-stage follow a distribution with mean $\mu$ and standard deviation $\sigma$. Let the total number of entiities be $N$, with the high-score subset containing $N_h$ entities and low-confidence subset containing $N_l$ entities. Denote the maximum score within the high-score subset as

$$S_{e_h} = \max_{i=1,...,N_h} s_i \tag{20}$$

Denote the maximum score within the low-score subset as

$$S_{e_l} = \max_{i=N_h+1,...,N} s_i \tag{21}$$

According to Eqn 19, then the expectation of the maximum of the two subsets satisfies

$$\mathbb{E}[S_{e_h}] \geq (\mu + \frac{\sigma}{N_h}) \cdot \frac{N_h}{N_h^2 + 1}$$
$$\mathbb{E}[S_{e_l}] \geq (\mu + \frac{\sigma}{N_l}) \cdot \frac{N_l}{N_l^2 + 1} \tag{22}$$

The gap between the two is

$$\begin{aligned}|\mathbb{E}[S_{e_h}] - \mathbb{E}[S_{e_l}]| &\geq \left|(\mu + \frac{\sigma}{N_h}) \cdot \frac{N_h}{N_h^2 + 1} - (\mu + \frac{\sigma}{N_l}) \cdot \frac{N_l}{N_l^2 + 1}\right| \\ &= \left|(\frac{N_h}{N_h^2 + 1} - \frac{N_l}{N_l^2 + 1}) \cdot \mu - (\frac{1}{N_h^2 + 1} - \frac{1}{N_l^2 + 1}) \cdot \sigma\right|\end{aligned} \tag{23}$$

In our implementation, $N_h$ is less than 10 and the number of entities in all datasets is more than 10000. Then, we have $\frac{N_l}{N_h} > 1000$, and the function $f(x) = \frac{x}{x^2+1}$ is monotonically decreasing for $x > 1$. Therefore,

$$\begin{aligned}|\mathbb{E}[S_{e_h}] - \mathbb{E}[S_{e_l}]| &\geq \left|(\frac{1}{N_h^2 + 1} - \frac{1}{N_l^2 + 1}) \cdot \sigma\right| \\ &\geq \left|(\frac{1}{N_h^2 + 1} - \frac{1}{(1000 N_h)^2 + 1}) \cdot \sigma\right| \\ &\approx \frac{1}{N_h^2 + 1} \cdot \sigma > 0.1 \cdot \sigma\end{aligned} \tag{24}$$

By Jensen's inequality, we have

$$\mathbb{E}[|S_{e_h} - S_{e_l}|] > |\mathbb{E}[S_{e_h}] - \mathbb{E}[S_{e_l}]| > 0.1 \cdot \sigma \tag{25}$$

We have demonstrated that, in our coarse-to-fine reasoning optimization, the expected score gap between the high-score and low-score subsets is at least 0.1 times the standard deviation. In comparision, other baseline methods (as shown in Figure 1) exhibit score gaps between correct and incorrect answers are typically less than $0.02\sigma$. This demonstrates that our optimization can amplify the score gap, thus mitigating over-smoothing.

$\square$

## A.3  Coarse-to-Fine Reasoning Optimization Improves the Quality of KG Reasoning.

*Proof.* In coarse-grained reasoning, the candidate entities are divided into two subsets, the high-score subset is denoted as $\mathcal{T}^{high}$. The highest-score entity in each subset, as computed by our proposed dual-pathway fusion model, is denoted as:

$$e_h = \underset{e \in \mathcal{T}^{high}}{\arg \max}\, s(e), \qquad e_l = \underset{e \notin \mathcal{T}^{high}}{\arg \max}\, s(e). \tag{26}$$

where the $s(\cdot)$ denotes score computing by dual-pathway fusion model. Let $P_\Delta$ denote the probability that the difference between $e_h$ and $e_l$ is less than or equal to $\Delta$, i.e. $P_\Delta = P(e_l - e_h \le \Delta)$.

Let $P$ and $P'$ denotes the probabilities of correctly identifying the answer with and without coarse-to-fine optimization, respectively. Let event $A$ denote that the HousE model assigns the ground-truth answer a score that ranks within the top-$k$ among all candidate entities, and event $B$ denote that our proposed dual-pathway model correctly infers the ground-truth answer. Therefore, the probability that coarse-to-fine reasoning accurately infers the correct answer is:

$$\begin{aligned}
P &= P_\Delta \cdot P(B \mid A) + (1 - P_\Delta) \cdot P' \\
&= P_\Delta \cdot P(B \mid A) + P' - P_\Delta \cdot P' \\
&= (P(B \mid A) - P') \cdot P_\Delta + P'
\end{aligned} \tag{27}$$

In the following, we compare the magnitude relationship between $P(B \mid A)$ and $P'$. $P(B \mid A)$ represents the probability of event $B$ occurring given that event $A$ has occurred. Specifically, the probability that the dual-pathway fusion model correctly infers the correct answer given that the correct answer is ranked within the high-score subset by coarse-grained reasoning. Evidently, the probability of the dual-pathway fusion model correctly inferring the correct answer is higher when the correct answer is already ranked within the high-score subset by coarse stage, compared to the unconditional probability of the dual-pathway fusion model's correct inference. This is because the high-score subset from coarse provides the dual-pathway fusion model with a more focused and promising subset.

Therefore, we can obtain that $P(B \mid A) \ge P'$ which leads to $P \ge P'$, thus demonstrating that the probability that coarse-to-fine reasoning optimization accurately infers the correct answer is more than the probability that dual-pathway fusion model without coarse-to-fine optimization correctly infers the correct answer. $\qquad \square$

# B  Time Complexity Computation

In this section, we provide details of time complexity computation in Section 3.1 and Section 3.2

## B.1  Time Complexity Computation of Dual-Pathway Global-Local Fusion Model.

**Time complexity of dual-pathway fusion model.**  We assume dual-pathway fusion model includs $L_m$ message passing layers and $L_t$ transformer layers. Here, $|\mathcal{V}|$ and $|\mathcal{E}|$ respectively denote the number of entities and triplets and $d$ is the dimension of entity representation. For each message passing layer, its time complexity is $\mathcal{O}(|\mathcal{E}|d + |\mathcal{V}d^2|)$. For each transformer layer, its time complexity is $\mathcal{O}(|\mathcal{V}d^2|)$. Because of message passing layer and transformer in parallel, the overall time complexity of our dual-pathway fusion model is $\mathcal{O}(\max(L_m(|\mathcal{E}|\,d + |\mathcal{V}|\,d^2), L_t\,|\mathcal{V}|\,d^2))$.

**Time complexity of one-pathway model.**  We assume one-pathway fusion model includs $L_m$ message passing layers and $L_t$ transformer layers. For each message passing layer, its time complexity is $\mathcal{O}(|\mathcal{E}|d + |\mathcal{V}d^2|)$. Here, $|\mathcal{V}|$ and $|\mathcal{E}|$ respectively denote the number of entities and triplets and $d$ is the dimension of entity representation. For each transformer layer, its time complexity is $\mathcal{O}(|\mathcal{V}d^2|)$. Because message passing layer and transformer is sequntial, the overall time complexity of our dual-pathway fusion model are $\mathcal{O}(L_m\,|\mathcal{E}|\,d + (L_m + L_t)\,|\mathcal{V}|\,d^2)$.

## B.2  Time Complexity Computation of Coarse-to-Fine Stage.

The coarse-to-fine reasoning stage has two additional operations of coarse-grained reasoning and sorting all entities compared to one-stage. In coarse-to-fine reasoning, parallel reasoning with coarse

Table 7: Transductive KG reasoning performance for DuetGraph, SimKGC and MoCoKGC on FB15k-237 and WN18RR. (The best results are bolded in red with a yellow highlight. )

| Value | FB15k-237 | | | WN18RR | | |
|---|---|---|---|---|---|---|
| | MRR | H@1 | H@10 | MRR | H@1 | H@10 |
| SimKGC[57] | 0.336 | 24.9 | 51.1 | 0.666 | 58.5 | **80.0** |
| MoCoKGC[65] | 0.391 | 29.6 | 43.1 | **0.742** | **66.5** | 79.2 |
| DuetGraph(ours) | **0.456** | **36.1** | **62.8** | 0.594 | 54.2 | 70.0 |

model and fine model. Moreover, the time complexity of the coarse model we employ is $\mathcal{O}(|\mathcal{E}|d)$ where $|\mathcal{E}|$ denote the number of triplets and $d$ is the dimension of entity representation. The time complexity of this sorting process is $\mathcal{O}(|\mathcal{V}|log|\mathcal{V}|)$, where $|\mathcal{V}|$ denote the number of entities. The one-stage reasoning only includes dual-pathway fusion model. Therefore, the overall time complexity of the coarse-to-fine reasoning stage is $\mathcal{O}(\max(L_m(|\mathcal{E}|\,d + |\mathcal{V}|\,d^2), L_t\,|\mathcal{V}|\,d^2) + |\mathcal{V}|\,log\,|\mathcal{V}|)$.

## C  Additional Baseline Discussion

### C.1  DuetGraph vs. Methods based on pre-trained language models.

We observe that language model-based reasoning methods such as SimKGC [57] and Mo-CoKGC [65] achieve unusually high results on WN18RR, but perform poorly on other datasets, as shown in Table 7. To better understand this phenomenon, we take these two methods as representative examples for further analysis. We note that WN18RR is derived from WordNet, a large lexical database of English that naturally encodes rich semantic relations between words. Pre-trained language models are well-suited to capturing such general semantic information, which may explain their strong performance on WN18RR. In contrast, FB15k-237 involves more domain-specific relational knowledge, which poses greater challenges for these models, leading to weaker performance (as shown in Table 7).

Additionally, we consider the possibility that the textual descriptions of entities in WN18RR may have appeared in the pretraining corpus of language models, potentially leading to data leakage. Therefore, we adopt the detection method proposed by [66] to estimate the proportion of WN18RR entity texts that are likely included in the pretraining data of the language model used by SimKGC and MoCoKGC (i.e., bert-base-uncased).

Specifically, for an entity text, select the $\epsilon$ of tokens with the lowest predicted probabilities from the language model. Then, compute the average log-likelihood of these low-probability tokens. If the average log-likelihood exceeds a certain threshold, we consider that the text is likely to have appeared in the language models pre-training data.

The detailed results are presented in Table 8. We observe that even under smaller $\epsilon$ (e.g.,10% and 20%) that means selecting the $\epsilon$ of tokens that are most difficult to be recognized by the language model, over 70% of the entity texts in WN18RR appear to be memorized by the language model, suggesting a significant potential for data leakage.

Table 8: Pre-training overlap rate under varying $\epsilon$, where $\epsilon$ represents the proportion of low probability tokens predicted by language model.

| $\epsilon$ | Pretraining Overlap Rate |
|---|---|
| 10.0% | 70.11% |
| 20.0% | 77.46% |
| 50.0% | 82.60% |
| 60.0% | 81.92% |

## C.2 Baseline Details.

In this section, we explain the reasons for not comparing with some baseline methods on certain datasets. SimKGC [57] requires additional textual information as part of its input data. Since the public repository does not provide textual information for some datasets (e.g., NELL-995), comparisons on those datasets are not conducted. DRUM [55] and A*Net [13] do not provide the specific parameters required to construct the inductive datasets as described in their papers. Therefore, they cannot be applied to certain datasets (e.g., NELL-995v1).

# D Experimental Details

## D.1 Transductive and Inductive Reasoning.

Following the formal definition in [67], transductive reasoning assumes that all test entities appear during training, while inductive reasoning handles completely unseen entities during testing. This difference is fundamental to evaluating model generalization capabilities.

Therefore, following the methodology in [68], we construct our inductive evaluation datasets by ensuring complete separation between training and test entities. This strict partitioning, where test entities are excluded from training, enables a reliable assessment of the model's generalization capability to unseen knowledge.

## D.2 Relation Prediction Task.

The relation prediction task $(h, ?, t)$ can indeed be transformed to fit our tail completion paradigm through the approach in [69]:

- **Scoring Mechanism:** For relation prediction, we fix head ($h$) and tail ($t$) entities, then score all candidate relations. For tail prediction, we fix the head ($h$) and relation ($r$), then score all candidate tails. Both tasks use the same underlying scoring function.

- **Implementation:** For relation prediction, we compute a score for each candidate relation and select the one with the highest score as the prediction.

This approach maintains fundamental consistency with tail entity prediction. While the surface-level structures differ, both tasks share the same underlying computational paradigm: evaluating possible completions against fixed components of the triple using a unified scoring mechanism.

## D.3 Dataset Statistics.

We conduct experiments on four knowledge graph reasoning datasets, and the statistics of these datasets are summarized in Table 9. The specific dataset details are as follows:

- The FB15k-237 [47] dataset is a subset of FB15k [35]. Toutanova and Chen [47] pointed out that WN18 and FB15k have a test set leakage problem. Therefore, they extracted FB15k-237 from FB15k.

- The WN18RR [48] dataset is a subset of WN18 [35]. All inverse relations in the WN18 dataset were removed by Dettmers et al. [48] to obtain the WN18RR dataset.

- NELL-995 [49] is a refined subset of the NELL knowledge base, curated for multi-hop reasoning tasks by filtering out low-value relations and retaining only the top 200 most frequent ones.

- YAGO3-10 [70] is a subset of YAGO3, containing 123,182 entities and 37 relations, where most relations provide descriptions of people. Some relationships have a hierarchical structure such as $playsFor$ or $actedIn$, while others induce logical patterns, like $isMarriedTo$.

Additionally, we perform experiments on three inductive knowledge graph reasoning datasets, each of which contains four different splits. The statistics of the inductive datasets are summarized in Table 10.

Table 9: Dataset Statistics for Tranductive Knowledge Graph Reasoning Datasets.

| Dataset | Relation | Entity | Triplet | | |
|---|---|---|---|---|---|
| | | | Train | Valid | Test |
| FB15k-237 [47] | 237 | 14,541 | 272,115 | 17,535 | 20,466 |
| WN18RR [48] | 11 | 40,943 | 86,835 | 3,034 | 3,134 |
| NELL-995 [49] | 200 | 74,536 | 149,678 | 543 | 2,818 |
| YAGO3-10 [70] | 37 | 123,182 | 1,079,040 | 5,000 | 5,000 |

Table 10: Dataset Statistics for Inductive Knowledge Graph Reasoning Datasets. In each split, one needs to infer Query triplets based Fact triplets.

| Dataset | Relation | Entity | Train | | | Valid | | | Test | | |
|---|---|---|---|---|---|---|---|---|---|---|---|
| | | | Entity | Query | Fact | Entity | Query | Fact | Entity | Query | Fact |
| FB15k-237 [47] | 180 (v1) | 1,594 | 1,594 | 4,245 | 4,245 | 1,594 | 489 | 4,245 | 1,093 | 205 | 1,993 |
| | 200 (v2) | 2,608 | 2,608 | 9,739 | 9,739 | 2,608 | 1,166 | 9,739 | 1,660 | 478 | 4,145 |
| | 215 (v3) | 3,668 | 3,668 | 17,986 | 17,986 | 3,668 | 2,194 | 17,986 | 2,501 | 865 | 7,406 |
| | 219 (v4) | 4,707 | 4,707 | 27,203 | 27,203 | 4,707 | 3,352 | 27,203 | 3,051 | 1,424 | 11,714 |
| WN18RR [48] | 9 (v1) | 2,746 | 2,746 | 5,410 | 5,410 | 2,746 | 630 | 5,410 | 922 | 188 | 1,618 |
| | 10 (v2) | 6,954 | 6,954 | 15,262 | 15,262 | 6,954 | 1,838 | 15,262 | 2,757 | 441 | 4,011 |
| | 9 (v3) | 12,078 | 12,078 | 25,901 | 25,901 | 12,078 | 3,097 | 25,901 | 5,084 | 605 | 6,327 |
| | 9 (v4) | 3,861 | 3,861 | 7,940 | 7,940 | 3,861 | 934 | 7,940 | 7,084 | 1,429 | 12,334 |
| NELL-995 [49] | 14 (v1) | 3,103 | 3,103 | 4,687 | 4,687 | 3,103 | 414 | 4,687 | 225 | 100 | 833 |
| | 86 (v2) | 2,564 | 2,564 | 15,262 | 8,219 | 2,564 | 922 | 8,219 | 2,086 | 476 | 4,586 |
| | 142 (v3) | 4,647 | 4,647 | 16,393 | 16,393 | 4,647 | 1,851 | 16,393 | 3,566 | 809 | 8,048 |
| | 76 (v4) | 2,092 | 2,092 | 7,546 | 7,546 | 2,092 | 876 | 7,546 | 2,795 | 7,073 | 731 |

## D.4 Hyperparameters Setup.

**Coarse-to-Fine reasoning model.** In the coarse-grained reasoning stage, we directly adopt existing models without any modifications to their original hyperparameter settings.

**Dual-Pathway fusion model.** For each dataset, we perform hyperparameter tuning on the validation set. We conduct grid search over the following hyperparameters:

- **Learning rate**: $\{10^{-4}, 5 \times 10^{-4}, 10^{-3}, 5 \times 10^{-3}, 10^{-2}\}$
- **Weight decay**: $\{10^{-5}, 10^{-4}\}$
- **Hidden dimension**: $\{16, 32, 64, 128\}$
- **Negative sampling size**: $\{128, 256, 512\}$
- **Message passing layers in input encoder**: $\{1, 2, 3\}$
- **Message passing layers in local pathway**: $\{1, 2, 3\}$
- **Transformer layers in global pathway**: $\{1, 2, 3\}$

In addition, we initialize the value of $\alpha$ randomly within the range (0, 1). Since we use the same random seed for all datasets, the initial value of $\alpha$ is identical across different datasets and is 0.549. And we report the range of $\alpha$ values observed during training across different datasets, as shown in the Table 11, Table 12 and Table 13. The results show that $\alpha$ consistently converges to a stable value during training, with negligible fluctuations afterward (less than 0.001). Across all datasets, $\alpha$ converges reliably as expected. Moreover, $\alpha$ remains below the theoretical threshold $\frac{\sigma_{\max}(\mathcal{M}_D) + \sigma_{\max}(\mathcal{M}_O)}{1 + \sigma_{\max}(\mathcal{M}_O)}$ as stated in Theorem 1.

**Coarse-to-Fine Optimization.** In the coarse-to-fine optimization, two key hyperparameters are involved: the number of entities in the high-confidence subset $k$, and the decision threshold $\Delta$.

We analyze the impact of the hyperparameter $k$, which denotes the number of entities in the high-score subset. We conduct experiments on the validation sets of all transductive datasets using our model. We set $k$ to $\{1, 2, 3, 4, 5, 6, 7, 8, 9, 10, 11, 12, 13, 14, 15\}$. See Table 15. We ultimately

Table 11: The range of $\alpha$ values observed during training on FB15k-237.

| Datasets | FB15k-237v1 | FB15k-237v2 | FB15k-237v3 | FB15k-237v4 |
|---|---|---|---|---|
| Theoretical threshold | 2.27 | 2.44 | 2.40 | 2.46 |
| $\alpha$ (Epoch=0) | 0.549 | 0.549 | 0.549 | 0.549 |
| $\alpha$ (Epoch=2) | 0.538 | 0.518 | 0.513 | 0.507 |
| $\alpha$ (Epoch=4) | 0.531 | 0.501 | 0.498 | 0.486 |
| $\alpha$ (Epoch=6) | 0.522 | 0.485 | 0.477 | 0.465 |
| $\alpha$ (Epoch=8) | 0.514 | 0.468 | 0.460 | 0.450 |
| $\alpha$ (Epoch=10) | 0.508 | 0.457 | 0.445 | 0.437 |
| $\alpha$ (Epoch=12) | 0.509 | 0.458 | 0.445 | 0.439 |
| $\alpha$ (Epoch=14) | 0.511 | 0.460 | 0.447 | 0.440 |
| $\alpha$ (Epoch=16) | 0.512 | 0.461 | 0.448 | 0.441 |
| $\alpha$ (Epoch=18) | 0.512 | 0.461 | 0.449 | 0.441 |
| $\alpha$ (Epoch=20) | 0.512 | 0.461 | 0.449 | 0.441 |

Table 12: The range of $\alpha$ values observed during training on WN18RR.

| Datasets | WN18RRv1 | WN18RRv2 | WN18RRv3 | WN18RRv4 |
|---|---|---|---|---|
| Theoretical threshold | 2.17 | 2.00 | 2.03 | 2.10 |
| $\alpha$ (Epoch=0) | 0.549 | 0.549 | 0.549 | 0.549 |
| $\alpha$ (Epoch=2) | 0.539 | 0.543 | 0.546 | 0.543 |
| $\alpha$ (Epoch=4) | 0.534 | 0.531 | 0.534 | 0.540 |
| $\alpha$ (Epoch=6) | 0.533 | 0.524 | 0.536 | 0.537 |
| $\alpha$ (Epoch=8) | 0.532 | 0.512 | 0.523 | 0.536 |
| $\alpha$ (Epoch=10) | 0.531 | 0.504 | 0.511 | 0.532 |
| $\alpha$ (Epoch=12) | 0.531 | 0.505 | 0.511 | 0.532 |
| $\alpha$ (Epoch=14) | 0.533 | 0.505 | 0.510 | 0.533 |
| $\alpha$ (Epoch=16) | 0.533 | 0.505 | 0.511 | 0.533 |
| $\alpha$ (Epoch=18) | 0.533 | 0.505 | 0.511 | 0.533 |
| $\alpha$ (Epoch=20) | 0.533 | 0.505 | 0.511 | 0.533 |

Table 13: The range of $\alpha$ values observed during training on NELL-995.

| Datasets | NELL-995v1 | NELL-995v2 | NELL-995v3 | NELL995v4 |
|---|---|---|---|---|
| Theoretical threshold | 2.49 | 2.58 | 2.54 | 2.42 |
| $\alpha$ (Epoch=0) | 0.549 | 0.549 | 0.549 | 0.549 |
| $\alpha$ (Epoch=2) | 0.544 | 0.536 | 0.526 | 0.536 |
| $\alpha$ (Epoch=4) | 0.539 | 0.532 | 0.496 | 0.516 |
| $\alpha$ (Epoch=6) | 0.523 | 0.525 | 0.474 | 0.500 |
| $\alpha$ (Epoch=8) | 0.514 | 0.517 | 0.447 | 0.487 |
| $\alpha$ (Epoch=10) | 0.514 | 0.509 | 0.424 | 0.477 |
| $\alpha$ (Epoch=12) | 0.514 | 0.509 | 0.425 | 0.475 |
| $\alpha$ (Epoch=14) | 0.512 | 0.510 | 0.425 | 0.476 |
| $\alpha$ (Epoch=16) | 0.512 | 0.510 | 0.426 | 0.476 |
| $\alpha$ (Epoch=18) | 0.512 | 0.510 | 0.427 | 0.476 |
| $\alpha$ (Epoch=20) | 0.512 | 0.510 | 0.427 | 0.476 |

select $k = 4$ for all the datasets, as these settings yield relatively high and stable results across three key metrics MRR, Hits@1, and Hits@10 rather than optimizing a single metric in isolation. We also conduct the same experiments on 4 inductive datasets (As shown in Figure 5).

Additionally, We analyze the impact of the hyperparameter, the decision threshold $\Delta$. We run experiments on all transductive datasets with our model. We set $\Delta$ to {0, 0.5, 1, 1.5, 2, 2.5, 3, 3.5, 4, 4.5, 5, 5.5, 6, 6.5, 7, 7.5, 8, 8.5, 9, 9.5, 10}. For FB15k-237 and WN18RR, we set $\Delta$ to 8 and set $\Delta$ to 5 for NELL-995 and YAGO3-10. The results are shown in Table 14.

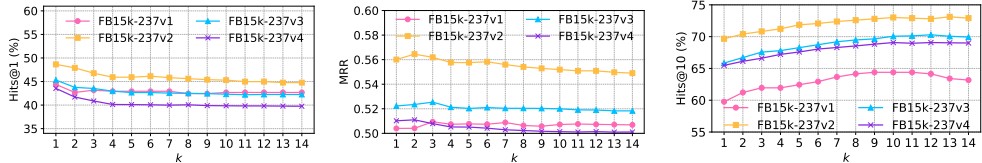

Figure 5: Effect of hyperparameter $k$ on the performance metrics of KG reasoning for different datasets (inductive).

Table 14: The results on transductive knowledge graph reasoning datasets with different $\Delta$.

| Value | FB15k-237 | | | WN18RR | | | NELL-995 | | | YAGO3-10 | | |
|---|---|---|---|---|---|---|---|---|---|---|---|---|
| | MRR | H@1 | H@10 | MRR | H@1 | H@10 | MRR | H@1 | H@10 | MRR | H@1 | H@10 |
| 0.0 | 0.411 | 31.9 | 58.0 | 0.581 | 52.5 | 68.7 | 0.553 | 48.1 | 66.5 | 0.600 | 53.0 | 72.3 |
| 0.5 | 0.423 | 33.0 | 59.2 | 0.584 | 53.0 | 68.9 | 0.572 | 50.1 | 69.3 | 0.613 | 54.3 | 73.0 |
| 1.0 | 0.436 | 34.1 | 60.6 | 0.588 | 53.4 | 69.0 | 0.579 | 50.8 | 69.8 | 0.619 | 54.8 | 73.5 |
| 1.5 | 0.443 | 34.7 | 61.3 | 0.589 | 53.6 | 69.0 | 0.585 | 51.4 | 70.5 | 0.623 | 55.2 | 74.0 |
| 2.0 | 0.447 | 35.2 | 61.7 | 0.590 | 53.7 | 69.1 | 0.589 | 51.7 | 71.1 | 0.626 | 55.5 | 74.4 |
| 2.5 | 0.448 | 35.3 | 61.7 | 0.591 | 53.8 | 69.1 | 0.590 | 51.8 | 71.2 | 0.628 | 55.7 | 74.6 |
| 3.0 | 0.450 | 35.5 | 61.8 | 0.591 | 53.8 | 69.1 | 0.592 | 52.0 | 71.2 | 0.630 | 55.8 | 74.7 |
| 3.5 | 0.451 | 35.6 | 61.9 | 0.591 | 53.8 | 69.2 | 0.592 | 52.1 | 71.2 | 0.631 | 56.0 | 74.8 |
| 4.0 | 0.452 | 35.7 | 62.0 | 0.591 | 53.9 | 69.2 | 0.592 | 52.1 | 71.2 | 0.632 | 55.1 | 74.8 |
| 4.5 | 0.452 | 35.7 | 62.1 | 0.591 | 53.9 | 69.2 | 0.592 | 52.1 | 71.2 | 0.632 | 56.1 | 74.9 |
| 5.0 | 0.453 | 35.8 | 62.2 | 0.592 | 53.9 | 69.2 | **0.593** | **52.2** | **71.2** | **0.632** | **56.1** | **74.9** |
| 5.5 | 0.453 | 35.8 | 62.2 | 0.592 | 53.9 | 69.2 | 0.593 | 52.2 | 71.1 | 0.632 | 56.1 | 74.9 |
| 6.0 | 0.454 | 35.9 | 62.2 | 0.592 | 53.9 | 69.2 | 0.593 | 52.2 | 71.1 | 0.632 | 56.1 | 74.8 |
| 6.5 | 0.454 | 35.9 | 62.3 | 0.592 | 53.9 | 69.2 | 0.592 | 52.2 | 71.1 | 0.632 | 56.1 | 74.8 |
| 7.0 | 0.455 | 36.0 | 62.4 | 0.592 | 54.0 | 69.2 | 0.592 | 52.2 | 71.0 | 0.632 | 56.1 | 74.9 |
| 7.5 | 0.456 | 36.1 | 62.5 | 0.593 | 54.2 | 69.9 | 0.592 | 52.2 | 71.0 | 0.632 | 56.1 | 74.9 |
| 8.0 | **0.456** | **36.1** | **62.8** | **0.594** | **54.0** | **70.0** | 0.592 | 52.1 | 71.0 | 0.632 | 56.1 | 74.9 |
| 8.5 | 0.457 | 36.1 | 62.7 | 0.593 | 54.0 | 69.9 | 0.592 | 52.1 | 69.9 | 0.632 | 56.0 | 74.9 |
| 9.0 | 0.456 | 35.9 | 62.6 | 0.593 | 54.0 | 69.9 | 0.592 | 52.1 | 69.9 | 0.632 | 56.0 | 74.9 |
| 9.5 | 0.454 | 35.9 | 62.6 | 0.593 | 54.0 | 69.2 | 0.592 | 52.1 | 69.9 | 0.632 | 56.0 | 74.9 |
| 10.0 | 0.454 | 35.9 | 62.2 | 0.593 | 54.1 | 69.3 | 0.592 | 52.1 | 69.9 | 0.632 | 56.0 | 74.9 |

**Computational Environment.** The experiments are conducted using Python 3.9.21, PyTorch 2.6.0, and CUDA 12.1, with an NVIDIA A100 80GB GPU.

### D.5 Random Initialization.

We run each model three times with different random seeds and report the mean results. We do not report the error bars because our model has very small errors with respect to random initialization. The standard deviations of the results are very small. For example, the standard deviations of MRR, H@1 and H@10 of DuetGraph are $9.8 \times 10^{-7}$, $5.6 \times 10^{-7}$ and $1.95 \times 10^{-5}$ on FB15k-237 dataset, respectively. On WN18RR dataset, the standard deviations of MRR, H@1 and H@10 of DuetGraph are $1.607 \times 10^{-6}$, $7.77 \times 10^{-6}$ and $8.94 \times 10^{-6}$, respectively. On NELL-995 dataset, the standard deviations of MRR, H@1 and H@10 of DuetGraph are $5.625 \times 10^{-7}$, $1.0 \times 10^{-8}$ and $1.89 \times 10^{-5}$, respectively. On YAGO3-10 dataset, the standard deviations of MRR, H@1 and H@10 of DuetGraph are $2.64 \times 10^{-6}$, $4.3 \times 10^{-7}$ and $2.5 \times 10^{-5}$, respectively. This indicates that our model is not sensitive to the random initialization.

### D.6 Ranking Protocol.

Following previous work [19], we conducted experiments on DuetGraph using the strictest ranking protocol $(m + n + 1)$, where $m$ is the number of entities with higher scores than the correct answer and $n$ is the number of entities that receive the same score as the correct answer. We also conducted experiments using a widely used but more lenient ranking protocols, namely $(m + 1)$ adopted in [71, 36, 72, 73], as shown in Table 16 and Table 17.

Based on the experimental results, we can draw the following conclusions.

Table 15: The results on transductive knowledge graph reasoning datasets with different $k$.

| Value | FB15k-237 | | | WN18RR | | | NELL-995 | | | YAGO3-10 | | |
|---|---|---|---|---|---|---|---|---|---|---|---|---|
| | MRR | H@1 | H@10 | MRR | H@1 | H@10 | MRR | H@1 | H@10 | MRR | H@1 | H@10 |
| 1 | 0.455 | 37.7 | 61.6 | 0.602 | 55.0 | 70.4 | 0.592 | 54.6 | 68.1 | 0.638 | 58.3 | 74.2 |
| 2 | 0.457 | 37.0 | 61.9 | 0.599 | 53.6 | 70.6 | 0.594 | 54.6 | 68.2 | 0.636 | 57.0 | 74.4 |
| 3 | 0.456 | 36.4 | 62.3 | 0.595 | 52.4 | 70.9 | 0.592 | 54.1 | 68.4 | 0.634 | 56.5 | 74.6 |
| 4 | 0.456 | 36.1 | 62.8 | 0.593 | 52.2 | 71.2 | 0.592 | 54.1 | 68.5 | 0.632 | 56.1 | 74.9 |
| 5 | 0.454 | 35.8 | 63.0 | 0.591 | 52.0 | 71.4 | 0.593 | 54.1 | 68.7 | 0.631 | 55.9 | 75.2 |
| 6 | 0.452 | 35.5 | 63.4 | 0.589 | 51.9 | 71.5 | 0.593 | 54.1 | 69.0 | 0.629 | 55.7 | 75.4 |
| 7 | 0.452 | 35.4 | 63.9 | 0.589 | 51.8 | 71.7 | 0.594 | 54.2 | 69.3 | 0.628 | 55.5 | 75.6 |
| 8 | 0.451 | 35.4 | 64.2 | 0.588 | 51.7 | 71.9 | 0.593 | 54.1 | 69.5 | 0.627 | 55.4 | 75.8 |
| 9 | 0.451 | 35.4 | 64.6 | 0.588 | 51.7 | 72.1 | 0.594 | 54.2 | 69.8 | 0.626 | 55.3 | 75.9 |
| 10 | 0.450 | 35.3 | 65.1 | 0.587 | 51.6 | 72.4 | 0.594 | 54.2 | 70.0 | 0.626 | 55.3 | 76.2 |
| 11 | 0.450 | 35.3 | 65.2 | 0.587 | 51.5 | 72.1 | 0.595 | 54.1 | 70.1 | 0.625 | 55.2 | 76.2 |
| 12 | 0.450 | 35.3 | 65.2 | 0.584 | 51.0 | 72.1 | 0.593 | 53.9 | 70.0 | 0.624 | 55.1 | 76.2 |
| 13 | 0.449 | 35.2 | 65.1 | 0.583 | 50.9 | 72.2 | 0.592 | 53.9 | 69.9 | 0.624 | 55.0 | 76.1 |
| 14 | 0.449 | 35.2 | 65.0 | 0.583 | 50.9 | 72.1 | 0.592 | 53.9 | 69.9 | 0.623 | 55.0 | 76.0 |
| 15 | 0.447 | 35.0 | 64.9 | 0.582 | 50.9 | 72.0 | 0.591 | 53.8 | 69.8 | 0.623 | 54.9 | 75.8 |

Table 16: Comparision of different under ranking protocols across four datasets.

| Method | FB15k-237 | | | WN18RR | | | NELL-995 | | | YAGO3-10 | | |
|---|---|---|---|---|---|---|---|---|---|---|---|---|
| | MRR | H@1 | H@10 | MRR | H@1 | H@10 | MRR | H@1 | H@10 | MRR | H@1 | H@10 |
| KnowFormer ($m+n+1$) | 0.430 | 34.3 | 60.8 | 0.579 | 52.8 | 68.7 | 0.566 | 50.2 | 67.5 | 0.615 | 54.7 | 73.4 |
| DuetGraph ($m+n+1$) | 0.453 | 36.1 | 62.4 | 0.593 | 54.2 | 69.9 | 0.590 | 52.1 | 71.2 | 0.631 | 56.1 | 74.8 |
| DuetGraph ($m+1$) | 0.456 | 36.1 | 62.8 | 0.594 | 54.2 | 70.0 | 0.593 | 52.2 | 71.2 | 0.632 | 56.1 | 74.9 |

- First, switching to the strictest ranking protocol has little impact on DuetGraphs quality. As shown in the tables above, when switching the ranking protocol from $(m+1)$ to $(m+n+1)$, DuetGraphs quality is virtually unaffected, with at most a 0.3% drop in MRR, a 0.1% drop in Hits@1, and a 0.4% drop in Hits@10. This is because, $n$ is 0 in almost all the cases. As shown in the table above, the number of test triplets impacted when replacing $(m+1)$ with $(m+n+1)$ is less than 1%.

- Second, DuetGraph achieves SOTA results even under the strictest ranking protocol. As shown above, it surpasses AnyBURL and KnowFormer across all datasets in MRR, Hits@1, and Hits@10. For the other baselines, as shown in Table 2 used the official code from their original papers, all of which employ protocols no stricter than $(m+n+1)$. As shown in [76], increasing the strictness of ranking protocol yields quality that is no higher (and often lower). Even under this tough setting, DuetGraph consistently outperforms all of them, providing strong evidence of its SOTA performance.

# E   More Experimental Results

## E.1   Model Size and Inference Time.

Despite employing a two-stage reasoning strategy, DuetGraph exhibits significantly superior inference efficiency compared to KnowFormer[19]. This advantage is primarily attributed to our innovative parallel processing architecture and a lightweight coarse-grained model selection mechanism. These components work synergistically, enabling DuetGraph to achieve this higher efficiency at a comparable model scale, as detailed in Table 18.

As shown in Table 18, the inference efficiency improvement is especially notable on large-scale knowledge graphs, such as YAGO3-10, where our method achieves a 34.2% increase in inference efficiency compared to the SOTA model KnowFormer[19].

## E.2   The difference between $\alpha$ and graph attention.

The main differences are reflected in the following three perspectives:

Table 17: Comparison of different ranking protocols across very large knowledge graphs. (Non-deterministic ranking [74, 63, 75] means that when two entities share the same score, their original order is preserved in the ranking. Compared with the $m + n + 1$ ranking protocol, this is more lenient.)

| Method | Wikidata5M | | | Freebase | | |
|---|---|---|---|---|---|---|
| | MRR | H@1 | H@10 | MRR | H@1 | H@10 |
| AnyBURL (Non-Deterministic Ranking) | 0.350 | 30.9 | 42.9 | 0.588 | 53.6 | 68.2 |
| KnowFormer ($m + n + 1$) | 0.332 | 26.7 | 46.3 | 0.684 | 65.7 | 73.6 |
| DuetGraph ($m + n + 1$) | 0.363 | 32.7 | 49.5 | 0.697 | 69.3 | 73.8 |
| DuetGraph ($m + 1$) | 0.363 | 32.7 | 49.5 | 0.699 | 69.4 | 73.9 |

Table 18: Comparison of Model Size and Inference Time across different datasets.

| Datasets | FB15k-237 | | WN18RR | | NELL-995 | | YAGO3-10 | |
|---|---|---|---|---|---|---|---|---|
| Metrics | Size (M) | Time (ms) | Size (M) | Time (ms) | Size (M) | Time (ms) | Size (M) | Time (ms) |
| DuetGraph | 6.5 (Coarse: 0.6, Fine: 5.9) | **347.07** | 1.3 (Coarse: 0.9, Fine: 0.4) | **292.49** | 7.2 (Coarse: 2.3, Fine: 4.9) | **261.32** | 3.4 (Coarse: 2.3, Fine: 1.1) | **597.22** |
| KnowFormer[19] | 6.1 | 499.71 | 0.4 | 392.40 | 5.2 | 360.40 | 1.0 | 905.22 |

- **Technical Role.**: As illustrated in Figure 2, the graph attention mechanism (Step 2) first learns global weights, which subsequently inform the learning of the control parameter $\alpha$ (Step 4). The attention weights serve as intermediate representations that enable $\alpha$ to effectively balance local and global information.

- **Theoretical Advantage.**: Introducing $\alpha$ allows for better fusion of the global weights captured by the attention mechanism (Step 2) and the local weights acquired via message passing (Step 3), which helps mitigate over-smoothing and enhances the models quality, as shown in Theorem 1.

- **Experimental Study.**: Incorporating $\alpha$ achieves better performance compared to using attention mechanism alone, as shown in Table 19.

Table 19: Results comparing the model with and without the $\alpha$ parameter.

| Metrics | FB15k-237 | | | WN18RR | | | NELL-995 | | | YAGO3-10 | | |
|---|---|---|---|---|---|---|---|---|---|---|---|---|
| | MRR | Hits@1 | Hits@10 | MRR | Hits@1 | Hits@10 | MRR | Hits@1 | Hits@10 | MRR | Hits@1 | Hits@10 |
| w/ $\alpha$ | **0.456** | **36.1** | **62.8** | **0.594** | **54.2** | **70.0** | **0.593** | **52.2** | **71.2** | **0.632** | **56.1** | **74.9** |
| only w/ attention | 0.445 | 35.1 | 61.2 | 0.584 | 54.1 | 69.0 | 0.582 | 51.0 | 70.3 | 0.617 | 54.2 | 73.7 |

# F   Limitations and Broader Impacts

## F.1   Limitations

Although DuetGraph has demonstrated its effectiveness in improving reasoning performance on several public benchmarks, many challenges remain to be addressed. For instance, its black-box decision process poses challenges for domains such as biomedicine, where expert interpretability and traceability are essential. Future work may incorporate explainability modules along with interactive visualization tools to help users understand the reasoning process of the model, thereby improving its trustworthiness and applicability in real-world scenarios such as clinical diagnosis and drug discovery [77].

## F.2   Broader Impacts

DuetGraph is a framework for knowledge graph reasoning that offers strong support for predicting missing information in real-world social networks. And DuetGraph holds great potential for accelerating discovery in biomedical domains, such as drug repurposing and disease-gene association prediction.

