# OpenReview forum: "DuetGraph: Coarse-to-Fine Knowledge Graph Reasoning with Dual-Pathway Global-Local Fusion"
_NeurIPS.cc/2025/Conference — NeurIPS 2025 poster_

### Official Review · Reviewer_hDcp · 2025-07-01

**Clarity:** 2
**Significance:** 2
**Originality:** 2
**Rating:** 4
**Confidence:** 4

**Summary:**

This paper presents DuetGraph, a dual-pathway architecture for knowledge graph reasoning that integrates local and global information via a coarse-to-fine strategy. Experiments show consistent gains over strong baselines across multiple benchmarks.

While the design is novel and results are promising, concerns remain. The inference efficiency and model size are not evaluated, despite the system’s complex two-stage structure. The motivation for the parallel fusion mechanism is unclear and not empirically supported. Notably, removing the coarse-to-fine module results in performance worse than KnowFormer, raising questions about the necessity of this design. Additionally, the paper contains several reference errors.

**Questions:**

- The proposed model combines both global and local knowledge graph information. However, KnowFormer also adopts a hybrid strategy. The paper does not clearly articulate the novelty or necessity of introducing a parallel structure in this context. According to the ablation study, removing the Coarse-to-Fine reasoning module leads to performance even worse than that of KnowFormer, which raises further questions about the justification and effectiveness of this structural design.

**Ethical Concerns:**

["NO or VERY MINOR ethics concerns only"]

**Final Justification:**

The performance of DuetGraph appears to be good, but I have identified some serious issues during the discussion period:

* All evaluation lacks **transparency and rigor** because the authors have used a different evaluation protocol compared to baselines. Although the authors have claimed that this difference does not have a high impact, all evaluations should be recomputed.
* Too many things should be revised: 1) all evaluation results (i.e., all tables and all figures for analysis); 2) more discussions on the evaluation protocol; 3) additional experiments.
* Limited novelty: after reviewing their code, I found that the global pathway comes from existing methods like KnowFormer and the local pathway comes from existing methods like RED-GNN. This work can be seen as an ensemble approach to pursuing high performance.

I remain a positive score because the authors have made many efforts in addressing my concerns.

**Limitations:**

yes

**Quality:**

3

**Strengths And Weaknesses:**

Strengths:

- The dual-pathway architecture is a valuable contribution that effectively combines local and global information, addressing a common limitation in existing models.
- The empirical results are impressive, showing significant improvements over baseline methods in various datasets, which indicates the practical utility of the proposed approach.

weaknesses:

- Despite the demonstrated training efficiency and time complexity analysis, I remain concerned about the inference efficiency and model size due to the algorithm's complex two-stage and parallel structure. This concern is not addressed experimentally in the paper.
- The motivation behind the parallel design for fusing global and local information is insufficiently explained. Moreover, this aspect is not supported or verified through targeted experiments.
- There are several errors in the references listed in the paper.

---

> ### Author Rebuttal · Authors · 2025-07-31
>
> > **W1: Despite the demonstrated training efficiency and time complexity analysis, I remain concerned about the inference efficiency and model size due to the algorithm's complex two-stage and parallel structure. This concern is not addressed experimentally in the paper.**
>
> **Response to W1**: We appreciate your thoughtful comments regarding efficiency and model size. We would like to address these points through both architectural design choices and empirical validation:
>
> - **Parallel Architecture Advantage:** Our design's parallel operation of message passing and attention modules during inference provides inherent efficiency gains compared to sequential one-pathway architectures. This parallelism enables faster inference while maintaining model performance.
>
> - **Lightweight Coarse Model Selection:** We intentionally employ efficient triplet-based models for the coarse stage, which: 1) achieves the best performance due to its architectural diversity [1] (**Table 4**); 2) maintains a model size comparable to baselines.
>
> - **Empirical Efficiency Validation:** Our comprehensive evaluation against KnowFormer [2] shows consistent advantages, as shown in the table below.
>
> | Datasets   | FB15k-237  |                       | WN18RR     |                       | NELL-995   |                       | YAGO3-10   |                       |
> |------------|------------|-----------------------|------------|-----------------------|------------|-----------------------|------------|-----------------------|
> | Metrics    | Model Size | Inference Time (One Batch) | Model Size | Inference Time (One Batch) | Model Size | Inference Time (One Batch) | Model Size | Inference Time (One Batch) |
> | DuetGraph  | 6.5M (Coase Model size: 0.6M, Fine Model Size: 5.9M) | **347.07ms**      | 1.3M (Coase Model size: 0.9M, Fine Model Size: 0.4M) | **292.49ms**     | 7.2M (Coase Model size: 2.3M, Fine Model Size: 4.9M) | **261.32ms**     | 3.4M (Coase Model size: 2.3M, Fine Model Size: 1.1M) | **597.22ms**     |
> | KnowFormer | 6.1M       | 499.71ms              | 0.4M       | 392.40ms              | 5.2M       | 360.40ms              | 1.0M       | 905.22ms              |
>
> The results show that the inference efficiency improvement is especially notable on large-scale knowledge graphs, such as YAGO3-10, where our method achieves a **34.2%** increase in inference efficiency compared to the SOTA model KnowFormer [2]. We will incorporate this analysis in the revised manuscript to better highlight these efficiency advantages.
>
>
>
> > **Q1: The proposed model combines both global and local knowledge graph information. However, KnowFormer also adopts a hybrid strategy. The paper does not clearly articulate the novelty or necessity of introducing a parallel structure in this context. According to the ablation study, removing the Coarse-to-Fine reasoning module leads to performance even worse than that of KnowFormer, which raises further questions about the justification and effectiveness of this structural design.**
> >
> > **W2: The motivation behind the parallel design for fusing global and local information is insufficiently explained. Moreover, this aspect is not supported or verified through targeted experiments.**
>
> **Response to Q1 and W2:** We appreciate your thoughtful questions regarding the parallel architecture design. We would like to clarify both the theoretical motivations and empirical advantages of our parallel design.
>
> **Theoretically**, as shown in **lines 43-58 (Section 1)** of the manuscript, the one-pathway architecture tends to aggravate the **over-smoothing** problem. In contrast, our parallel design alleviates over-smoothing by increasing the score gap, thereby improving model quality, as demonstrated in **Theorem 1**.
>
> **Experimentally**, we would like to clarify that the number of training epochs differs between the results for w/o Coarse-to-Fine reasoning (**Table 3** of the manuscript) and KnowFormer [2] (**Tables 2 and 4** of the manuscript). Specifically, the ablation experiments without Coarse-to-Fine reasoning were conducted with 15 epochs, whereas KnowFormer [2] was trained for 20 epochs.
>
> This is because DuetGraph converged in only 15 epochs, while other baselines (e.g., KnowFormer [2]) converged in 20 epochs. Therefore, to ensure **a fair comparison**, we trained all models (including DuetGraph) for 20 epochs, as shown in **Table 2** of the manuscript.
>
> In the ablation experiments (as shown in **Table 3** of the manuscript), we trained the model without Coarse-to-Fine reasoning for 15 epochs, matching the number of epochs required for DuetGraph to reach convergence, to ensure a **fair and consistent** comparison in the ablation experiments.
>
> To ensure a **rigorous and fair comparison** between the model without Coarse-to-Fine reasoning and KnowFormer [2], we additionally conducted ablation studies for both models under the same number of training epochs (i.e., 20 epochs), as shown in the tables below.
>
>
> |           Dataset            | FB15k-237 |          |          |  WN18RR   |          |          | NELL-995  |          |          | YAGO3-10  |          |          |
> | :--------------------------: | :-------: | :------: | :------: | :-------: | :------: | :------: | :-------: | :------: | :------: | :-------: | :------: | :------: |
> |           Metrics            |    MRR    |  Hits@1  | Hits@10  |    MRR    |  Hits@1  | Hits@10  |    MRR    |  Hits@1  | Hits@10  |    MRR    |  Hits@1  | Hits@10  |
> | w/o Coarse-to-Fine reasoning | **0.437** | **34.8** | **61.1** | **0.580** | **53.0** | **68.9** | **0.567** | **50.5** | **67.7** | **0.616** | **54.8** | **73.5** |
> |          KnowFormer          |   0.435   |   34.3   |   60.8   |   0.579   |   52.8   |   68.7   |   0.566   |   50.2   |   67.5   |   0.615   |   54.7   |   73.4   |
>
> | Dataset                        | FB15k-237 | WN18RR   | NELL-995  | YAGO3-10   |
> |:------------------------------:|:---------:|:--------:|:---------:|:----------:|
> | Metrics                        | One Epoch Time | One Epoch Time | One Epoch Time | One Epoch Time |
> | w/o Coarse-to-Fine reasoning   | **3,026.8s**   | **2,968.2s**   | **12,037.9s**  | **206,919.1s** |
> | KnowFormer                     | 3,471.6s       | 3,026.4s       | 12,329.2s      | 242,313.2s     |
>
> The results demonstrate that the parallel design not only outperforms the SOTA KnowFormer [2] in **quality** because of **alleviating over-smoothing**, but is also more **efficient** in terms of computational cost, including average epoch runtime cost. This is because parallel architectures, which are designed to leverage **parallel strategies**, are **more efficient** compared to sequential one-pathway architectures.
>
> Following your advice, we will include these experimental results and analysis in the revised manuscript.
>
> > **W3: There are several errors in the references listed in the paper.**
>
> **Response to W3**: Thanks for pointing these out. We will correct all the errors about references in the revised manuscript.
>
> ----
> **Reference**:
>
> [1] Ortega et al. Diversity and Generalization in Neural Network Ensembles. AISTATS 2022
>
> [2] Liu et al. KnowFormer: Revisiting Transformers for Knowledge Graph Reasoning. ICML 2024

---

> > ### Comment · Reviewer_hDcp · 2025-08-06
> >
> > Thanks for your detailed responses. My biggest concern still remains: the efficiency and scalability of this work are very limited. The authors fail to conduct experiments on very large KGs such as Wikidata5M [1] and Freebase [2], where some rule-based baselines such as AnyBURL [3] can be run on these large KGs within 3 hours. Besides, AnyBURL can achieve comparable performance as DuetGraph in YAGO3-10 by just spending 1000 seconds without using any contextual information. I recommend that the authors should discuss these rule-based baselines in detail.
> >
> > Refs:
> >
> > [1] Kepler: A unified model for knowledge embedding and pre-trained language representation. In TACL.
> >
> > [2] Parallel training of knowledge graph embedding models: a comparison of techniques. In VLDB.
> >
> > [3] Anytime bottom-up rule learning for large-scale knowledge graph completion. In VLDBJ.

---

> > > ### Author Response · Authors · 2025-08-06
> > >
> > > Thank you for your timely feedback on the comparison with rule-based baselines such as AnyBURL [1]. To clarify, we compare DuetGraph and AnyBURL from four perspectives.
> > >
> > > **First, although AnyBURL has higher learning efficiency, its quality is not as good as that of DuetGraph.**
> > >
> > > Rule-based methods (e.g., AnyBURL) focus on learning the rules, achieving high learning efficiency by employing **system-level** optimizations (e.g., using sampling to obtain random paths for rule mining and leveraging multithreading to accelerate this process). In contrast, the embedding-based methods (e.g, DuetGraph) focus on learning the embeddings (or representations) of KGs. These embedding-based methods are generalizable and transferable [2, 3], and they typically achieve **better quality** on downstream tasks.
> > >
> > > For example, as shown in the table below, DuetGraph achieves better quality than AnyBURL. Compared to AnyBURL, DuetGraph achieved a **6.7%** improvement in **MRR**, a **6.4%** improvement in **H@1**, and a **6%** improvement in **H@10**. The experimental settings are consistent with those of AnyBURL, i.e., DuetGraph also does not use any contextual information (as mentioned in **lines 263, Section 4.2** of the manuscript).
> > >
> > > |Method (KG: **YAGO3-10**)|MRR|H@1|H@10|Learning|Inference|Environment|
> > > |-|-|-|-|-|-|-|
> > > |AnyBURL [1] (Rule-based)|0.565|49.7|68.9|1,000s|221s|Two Intel(R) Xeon(R) CPU E5-2640 v4 @ 2.40GHz cores|
> > > |KnowFormer [4] (Embedding-based)|0.615|54.7|73.4|136h|755s|One A100 (80GB)|
> > > |DuetGraph (Embedding-based)|**0.632**|**56.1**|**74.9**|109h|505s|One A100 (80GB)|
> > >
> > > **Second, inference time—which is crucial for practical deployment—is comparable between DuetGraph and AnyBURL.**
> > >
> > > Although DuetGraph generally requires more learning time, the pre-learning process is performed **only once**, whereas inference is conducted **multiple times**. Therefore, inference time is more critical for practical deployment. As shown in the table above, our method achieves inference speeds of the **same order of magnitude** as AnyBURL. This is mainly because, due to vectorized operations, many candidates can be ranked simultaneously by GPUs [1]. What is more, further increasing GPU resources can further boost inference efficiency.
> > >
> > > **Third, DuetGraph’s learning efficiency can also be improved using system-level optimizations, similar to those in AnyBURL.**
> > >
> > > We clarify that **system-level** optimizations (e.g., **sampling** and **multithreading** in AnyBURL) are **orthogonal** to our DuetGraph design.  DuetGraph can further increase its efficiency by applying **system-level** optimizations to its **message passing** and **attention** modules. For example, in the message passing component, we can apply graph partitioning strategies [5, 6] to reduce redundant data loading, thereby optimizing I/O operations and improving training efficiency. In the attention component, IO-aware exact attention optimizations [7, 8] can be utilized to minimize memory reads and writes, further enhancing training efficiency. Therefore, DuetGraph’s learning efficiency can be further improved by incorporating these **system-level** optimizations.
> > >
> > > **Fourth, in response to your suggestion, we are currently running experiments on the Wikidata5M [9] and Freebase [10] datasets used in AnyBURL and will share the results shortly.**
> > >
> > > Following the experimental setup of the recent SOTA embedding-based methods (e.g., KnowFormer), we have conducted experiments on the large-scale YAGO3-10 dataset. Following your suggestion, we are currently conducting experiments on other large-scale datasets, such as Wikidata5M and Freebase. **We will report the results to you as soon as they become available.**
> > >
> > > Overall, the embedding-based methods (e.g., DuetGraph) and rule-based methods (e.g., AnyBURL) represent two distinct approaches to knowledge graph reasoning, each offering unique advantages. Following your suggestions, we will cite these papers and include these discussions about the rule-based baselines (e.g., AnyBURL) in the revised version of the manuscript.
> > >
> > > ---
> > >
> > > Refs:
> > >
> > > [1] Anytime bottom-up rule learning for large-scale knowledge graph completion. VLDBJ
> > >
> > > [2] Improving Visual Relationship Detection Using Semantic Modeling of Scene Descriptions. ISWC
> > >
> > > [3] Multilingual Knowledge Graph Embeddings for Cross-lingual Knowledge Alignment. IJCAI
> > >
> > > [4] KnowFormer: Revisiting Transformers for Knowledge Graph Reasoning. ICML
> > >
> > > [5] Capsule: An Out-of-Core Training Mechanism for Colossal GNNs. Proc. ACM Manag. Data
> > >
> > > [6] MariusGNN: Resource-Efficient Out-of-Core Training of Graph Neural Networks. Eurosys
> > >
> > > [7] FlashAttention: Fast and Memory-Efficient Exact Attention with IO-Awareness. NeurIPS
> > >
> > > [8] FlashAttention-2: Faster Attention with Better Parallelism and Work Partitioning. ICLR
> > >
> > > [9] Kepler: A unified model for knowledge embedding and pre-trained language representation. TACL
> > >
> > > [10] Parallel training of knowledge graph embedding models: a comparison of techniques. VLDB

---

> > > > ### Comment · Reviewer_hDcp · 2025-08-07
> > > >
> > > > Thanks for your further responses. I will keep my positive score. I just want to know how much time and GPU memory it will take for DuetGraph on large KGs such as Freebase. Is 80GB enough? It would be nice if the new experiments on large KGs could be included in the revised version.

---

> > > > > ### Author Response · Authors · 2025-08-08
> > > > >
> > > > > **Follow-up:**
> > > > > Thank you for maintaining your positive score for our paper. Following your suggestion, we have now completed the experiments on the **Wikidata5M** and **Freebase** datasets, as previously mentioned. The results are summarized below.
> > > > >
> > > > > |Method (KG: **Wikidata5M**)|MRR|H@1|H@10|Learning|Inference|Environment|Peak GPU memory usage|
> > > > > |-|-|-|-|-|-|-|-|
> > > > > |AnyBURL (Rule-based)|0.350|30.9|42.9|10,000s|4,462s|Two Intel(R) Xeon(R) Platinum 8358 CPU @ 2.60GHz|N/A|
> > > > > |KnowFormer (Embedding-based)|0.332|26.7|46.3|31,436s|105s|One A100 (80GB)|28.09GB|
> > > > > |DuetGraph (Embedding-based)|**0.363**|**32.7**|**49.5**|28,866s|**80s**|One A100 (80GB)|**25.73GB**|
> > > > >
> > > > > | Method (KG: **Freebase**)|MRR|H@1|H@10|Learning|Inference|Environment|Peak GPU memory usage|
> > > > > |-|-|-|-|-|-|-|-|
> > > > > |AnyBURL (Rule-based)|0.588|53.6|68.2|10,000s|3,142s|Two Intel(R) Xeon(R) Platinum 8358 CPU @ 2.60GHz|N/A|
> > > > > |KnowFormer (Embedding-based)| 0.684 | 65.7 | 73.6 | 32,109s | 176s |One A100 (80GB)|62.92GB|
> > > > > |DuetGraph (Embedding-based)  | **0.699** | **69.4** | **73.9** | 30,158s | **141s** | One A100 (80GB)|**57.49GB**|
> > > > >
> > > > > - It is worth noting that for the **Wikidata5M** and **Freebase** datasets, the AnyBURL paper [1] does not explicitly specify under which data split the results in **Table 2** of [1] were obtained. For a fair comparison, we use the same data split as in [2]: for **Freebase**, **304,727,650** training samples, **10,000** validation samples, and **10,000** test samples; and for **Wikidata5M**, **20,614,279** training samples, **5,163** validation samples, and **5,133** test samples. The hyperparameters of **AnyBURL** are set according to the values recommended by the authors in their original publication [1]. The experiments are conducted using these data splits.
> > > > >
> > > > > **Quality:** It can be concluded that **DuetGraph consistently outperforms AnyBURL in terms of model quality** on both the Wikidata5M and Freebase datasets by using **one A100 (80GB)**.
> > > > >
> > > > > - On **Wikidata5M**, DuetGraph achieves a **1.3% improvement in MRR**, a **1.8% improvement in H@1**, and a **6.6% improvement in H@10** compared to AnyBURL.
> > > > >
> > > > > - On **Freebase**, DuetGraph achieves an **11.1% improvement in MRR**, a **15.8% improvement in H@1**, and a **5.7% improvement in H@10** compared to AnyBURL.
> > > > >
> > > > > **Efficiency:** To enable a **fairer** comparison with AnyBURL in terms of learning and inference time, we applied similar **system-level** optimization techniques (e.g., **sampling**), as adopted by AnyBURL, to both DuetGraph and KnowFormer.
> > > > >
> > > > > As shown in the tables above, DuetGraph achieves **SOTA quality performance** with a learning time in **the same order of magnitude** as AnyBURL, and it demonstrates a **significant reduction** in inference time compared to AnyBURL.
> > > > >
> > > > > Specifically, on Wikidata5M, DuetGraph achieves an inference speed that is **55.8 times faster** than AnyBURL (**80s vs. 4,462s**). On Freebase, DuetGraph achieves an inference speed that is **22.3 times faster** than AnyBURL (**141s vs. 3,142s**).
> > > > >
> > > > > We sincerely appreciate your valuable suggestion, which enabled us to further validate our DuetGraph on these large-scale benchmarks and demonstrated its **advantages** on **larger** datasets. Following your suggestion, we will include the new experiments on large KGs in the revised version of our manuscript to further strengthen our contributions.
> > > > >
> > > > > ---
> > > > >
> > > > > Refs:
> > > > >
> > > > > [1] Anytime bottom-up rule learning for large-scale knowledge graph completion. VLDBJ
> > > > >
> > > > > [2] Parallel training of knowledge graph embedding models: a comparison of techniques. VLDB

---

> > > > > > ### Comment · Reviewer_hDcp · 2025-08-08
> > > > > >
> > > > > > Thank you for your detailed responses. Initially, I just doubt the efficiency and scalability of **DuetGraph**. However, after reading the authors' rebuttal, I was quite impressed by the reported performance gains (e.g., a 15.8% improvement in H\@1), which are indeed remarkable.
> > > > > >
> > > > > > That said, after carefully reviewing both the manuscript and the submitted code, I have identified a serious issue with the ranking protocol used in the evaluation of DuetGraph. Specifically, in Lines 114–122 of the file `coarse_to_fine_evaluate.py`, the rank of the correct answer is computed in the following way:
> > > > > >
> > > > > > ```python
> > > > > > ranks_final = torch.where(
> > > > > >         condition1,
> > > > > >         (top100_score > answer_score).sum(dim=1) + 1,          # At this point the correct answer is successfully narrowed down to the topk, and only the rank of the correct answer in the topk is calculated
> > > > > >         torch.where(
> > > > > >             condition2,
> > > > > >             (other_score > answer_score).sum(dim=1) + 1,        # At this point, the correct answer has been successfully narrowed down to the non-topk, and only the rank of the correct answer in the non-topk is calculated
> > > > > >             ((score_fine > answer_score) & (~mask)).sum(dim=1) +1 # At this point it did not succeed in narrowing down the correct answer to the subspace and calculating its ranking among all entities
> > > > > >         )
> > > > > >     )
> > > > > > ```
> > > > > >
> > > > > > This ranking method—computing the rank as *(m + 1)*, where *m* is the number of entities with higher scores than the correct answer—has been previously pointed out as problematic in RNNLogic, which is also included in your comparisons.
> > > > > >
> > > > > > The core issue lies in how tied scores are treated: if all entities receive the same score, the rank becomes 1, which overestimates the model's performance. As noted in RNNLogic, a more appropriate approach is to account for ties explicitly. In particular:
> > > > > >
> > > > > > * **RNNLogic** computes the rank as *(m + (n + 1)/2)*, where *n* is the number of entities that receive the **same** score as the correct answer.
> > > > > > * **KnowFormer**, which is arguably the most relevant baseline for your work, adopts a stricter formulation: *(m + n + 1)*.
> > > > > >
> > > > > > I strongly recommend that the authors adopt the same rank computation method as used in KnowFormer to ensure a **fair and consistent comparison** across methods.
> > > > > >
> > > > > > Given this issue, I believe all evaluations should be **recomputed** using the corrected ranking protocol. Additionally, a detailed discussion of the impact of this change should be included in the revised version to maintain transparency and rigor.

---

> ### Author Response · Authors · 2025-08-08
>
> Thank you for your detailed comments. Following the widely used ranking protocol adopted in [1–4], we employ the **(m+1)** ranking protocol in our experiments.
>
> Following your suggestion, we conducted experiments on DuetGraph using the **strictest** ranking protocol **(m+n+1)** (i.e., the one used in KnowFormer), and the results are shown in the tables below.
>
> |Dataset|FB15K-237|||WN18RR|||NELL-995|||YAGO3-10|||
> |-|-|-|-|-|-|-|-|-|-|-|-|-|
> |Metrics|MRR|Hits@1|Hits@10|MRR|Hits@1|Hits@10|MRR|Hits@1|Hits@10|MRR|Hits@1|Hits@10|
> |KnowFormer (m+n+1)|0.430|34.3|60.8|0.579|52.8|68.7|0.566|50.2|67.5|0.615|54.7| 73.4 |
> |DuetGraph (m+n+1)|**0.453 (-0.003)**|**36.1 (-0.0)**|**62.4 (-0.4)**|**0.593 (-0.001)**|**54.2 (-0.0)**|**69.9 (-0.1)**|**0.590 (-0.003)**|**52.1 (-0.1)**|**71.2 (-0.0)**|**0.631 (-0.001)**| **56.1 (-0.0)** |**74.8 (-0.1)**|
> |DuetGraph (m+1)|0.456|36.1|62.8|0.594|54.2|70.0|0.593|52.2|71.2|0.632|56.1|74.9|
>
> |Method (KG: Wikidata5M)|MRR|H@1|H@10|
> |-|-|-|-|
> |AnyBURL (Non-Deterministic Ranking [5-7])|0.350|30.9|42.9|
> |KnowFormer (m+n+1)|0.332|26.7|46.3|
> |DuetGraph (m+n+1)|**0.363 (-0.0)**|**32.7 (-0.0)**|**49.5 (-0.0)**|
> |DuetGraph (m+1)|0.363|32.7|49.5|
>
>
> | Method (KG: Freebase)|MRR|H@1|H@10|
> |-|-|-|-|
> |AnyBURL (Non-Deterministic Rank [5-7])|0.588|53.6|68.2|
> |KnowFormer (m+n+1)|0.684|65.7|73.6|
> |DuetGraph (m+n+1)|**0.697 (-0.002)**|**69.3 (-0.1)**|**73.8 (-0.1)**|
> |DuetGraph (m+1)|0.699|69.4|73.9|
> > Non-deterministic ranking  [5-7] means that when two entities share the same score, their original order is preserved in the ranking. Compared with the m+n+1 ranking protocol, this is **more lenient**.
>
> | Dataset | Total test triplets | The number of test triplets impacted when replacing (m+1) with (m+n+1) (DuetGraph) |
> |-|-|-|
> |FB15K-237|20,466|200 (0.98%)|
> |NELL-985|2,818|25 (0.89%)|
> |WN18RR|3,134|20 (0.64%)|
> |YAGO3-10|5,000|16 (0.98%)|
> |Wikidata5M|5,133|4 (0.07%)|
> |Freebase|10,000|86 (0.86%)|
>
> Based on the experimental results, we can draw the following conclusions.
>
> - **Frist, switching to the strictest ranking protocol has little impact on DuetGraph’s quality**. As shown in the tables above, when switching the ranking protocol from (**m+1**) to (**m+n+1**), DuetGraph’s quality is virtually unaffected, with **at most** a **0.003** drop in **MRR**, a **0.1** drop in **Hits@1**, and a **0.4** drop in **Hits@10**. This is because, **n** is **0** in almost all the cases . As shown in the table above, the number of test triplets impacted when replacing **(m+1)** with **(m+n+1)** is less than **1%**.
>
> - **Second, DuetGraph achieves SOTA results even under the strictest ranking protocol.** As shown above, it **surpasses** AnyBURL and KnowFormer across **all datasets** in **MRR**, **Hits@1**, and **Hits@10**. For the other baselines, as shown in **Table 2** of the manuscript, we used the **official code** from their original papers, all of which employ protocols **no stricter than (m+n+1)**.  As shown in [8], increasing the strictness of the ranking protocol yields quality that is no higher (and often lower). Even under this **tough** setting, DuetGraph consistently **outperforms** all of them, providing strong evidence of its **SOTA** performance.
>
> Following you suggestion, we will include these results and the discussion in the revised version of the manuscript. Thank you for reviewing our code and for the valuable feedback. This further demonstrates that DuetGraph delivers consistent **SOTA performance** under **the strictest** ranking protocol.
>
> ---
>
> Refs:
>
> [1] Pykg2vec: A Python Library for Knowledge Graph Embedding. J. Mach. Learn. Res.
>
> [2] Semi-Supervised Entity Alignment via Knowledge Graph Embedding with Awareness of Degree Difference. WWW
>
> [3] SimplE Embedding for Link Prediction in Knowledge Graphs. NeurIPS
>
> [4] Complex Embeddings for Simple Link Prediction. ICML
>
> [5] Multi-relational Poincaré Graph Embeddings. NeurIPS
>
> [6] Anytime bottom-up rule learning for large-scale knowledge graph completion. VLDBJ
>
> [7] On the ambiguity of rank-based evaluation of entity alignment or link prediction methods. arXiv
>
> [8] A Re-evaluation of Knowledge Graph Completion Methods. ACL

---

> > ### Comment · Reviewer_hDcp · 2025-08-09
> >
> > Thanks for your effort in addressing my concerns. I don't have further questions.

---

> > > ### Author Response · Authors · 2025-08-09
> > >
> > > Thank you very much for your time and effort in reviewing our paper, and even our code. If you are satisfied with our response, would you kindly consider bumping up your score?
> > >
> > > Thank you for your hard work and support.
> > >
> > > Best Regards,
> > >
> > > The authors of Paper **DuetGraph: Coarse-to-Fine Knowledge Graph Reasoning with Dual-Pathway Global-Local Fusion**

---

> > > ### Author Response · Authors · 2025-08-09
> > >
> > > Dear Reviewer hDcp,
> > >
> > > We hope this message finds you well. Despite acknowledging our clarifications, we noticed that you lowered the scores from **Clarity: 3 (good)**, **Significance: 4 (excellent)**, and **Originality: 3 (good)** to **Clarity: 2 (fair)**, **Significance: 2 (fair)**, and **Originality: 2 (fair)**.
> > >
> > > Could you clarify which aspects of our rebuttal were inadequate or unclear?
> > >
> > > Thank you.

---

> > > > ### Comment · Reviewer_hDcp · 2025-08-09
> > > >
> > > > I have updated my detailed evaluation because I have some new understandings of this work. My overall score is still positive. I think my updated evaluation is fair, considering both my initial evaluation and the further discussion. Good luck.

---

### Official Review · Reviewer_CuGL · 2025-07-02

**Clarity:** 3
**Significance:** 2
**Originality:** 2
**Rating:** 4
**Confidence:** 4

**Summary:**

This paper introduces DuetGraph, a new method for knowledge graph reasoning that solves the over-smoothing problem in deep models. The authors point out two main issues: combining local and global reasoning in one path causes over-smoothing, and single-stage models lack enough reasoning power. DuetGraph uses a dual-pathway design to separate local and global reasoning and combines their results with a learnable weight. It also uses a coarse-to-fine strategy, where a simple model first selects candidates, and a stronger model refines the results. A score gap-based method helps choose the final answer. The paper includes theory showing DuetGraph reduces over-smoothing better than older models, and experiments prove it achieves strong performance on multiple datasets.

**Questions:**

* Q1: In Line 81, the authors state that other KG completion tasks can be reformulated as tail entity completion. Could the authors explain how the relation prediction task (h, ?, t), which is structurally distinct, can be transformed into tail entity prediction task?
* Q2: The manuscript introduces the hyperparameter ∆ in the coarse-to-fine reasoning stage. Could the authors clarify how ∆ is set in the experiments? Additionally, it would be valuable to include an ablation study analyzing the sensitivity of model performance to different values of ∆, to better understand its impact.
* Q3: The coarse ranking model plays a crucial role in the DuetGraph framework. Could the authors provide an ablation study or analysis to quantify how much the coarse ranking component contributes to the overall performance?

**Ethical Concerns:**

["NO or VERY MINOR ethics concerns only"]

**Final Justification:**

Regarding Q1, the explanation on reformulating relation prediction into the tail entity completion paradigm is clear. The clarification that both tasks rely on the same scoring mechanism—fixing two elements and ranking the third—is convincing and consistent with prior approaches.

For Q2/W1, the discussion on the hyperparameter ∆ is now sufficiently addressed. The authors provided a clear grid search strategy and included a detailed ablation analysis demonstrating the model’s robustness to different ∆ values. The empirical results show stable improvements and low sensitivity, which alleviates the initial concern.

Regarding Q3/W2, the ablation study and architectural comparisons in Tables 3 and 4 effectively demonstrate the impact of the coarse ranking module. The quantitative degradation observed when removing the component, along with the justification for using the triplet-based architecture, provide strong support for its importance in the framework.

Overall, the author responses have resolved my concerns. I have no further questions at this point. I would like to keep my score.

**Limitations:**

Yes

**Quality:**

3

**Strengths And Weaknesses:**

# Strengths

* The paper proposes a novel dual-pathway architecture that decouples local (message passing) and global (transformer) reasoning. This design mitigates over-smoothing and enables parallel computation, leading to both improved performance and training efficiency.
* The paper provides rigorous theoretical analysis, including score gap bounds and convergence behavior, to support the proposed architecture and its claimed benefits.
* The experiments cover diverse datasets, include key ablation studies, and provide comparison on LLM based methods (performance and data leakage), offering solid empirical support for the proposed method.

# Weaknesses

* The hyperparameter ∆ used in the coarse-to-fine stage appears important to the final decision, yet its selection strategy is not discussed. There is no ablation study to show its impact on performance.
* The contribution of the coarse ranking model to overall performance is unclear. An ablation experiment isolating the effect of modifying the coarse stage would strengthen the claims.

---

> ### Author Rebuttal · Authors · 2025-07-31
>
> >**Q1: In Line 81, the authors state that other KG completion tasks can be reformulated as tail entity completion. Could the authors explain how the relation prediction task (h, ?, t), which is structurally distinct, can be transformed into the tail entity prediction task?**
>
> **Response to Q1**: Thanks for your comments. The relation prediction task (h, ?, t) can indeed be transformed to fit our tail completion paradigm through the approach in [1]:
>
> - **Scoring Mechanism:** For relation prediction, we fix head (h) and tail (t) entities, then score all candidate relations. For tail prediction, we fix the head (h) and relation (r), then score all candidate tails. Both tasks use the same underlying scoring function.
>
> - **Implementation:** For relation prediction, we compute a score for each candidate relation and select the one with the highest score as the prediction.
>
> This approach maintains fundamental consistency with tail entity prediction. While the surface-level structures differ, both tasks share the same underlying computational paradigm: evaluating possible completions against fixed components of the triple using a unified scoring mechanism.
>
> >**Q2: The manuscript introduces the hyperparameter ∆ in the coarse-to-fine reasoning stage. Could the authors clarify how ∆ is set in the experiments? Additionally, it would be valuable to include an ablation study analyzing the sensitivity of model performance to different values of ∆, to better understand its impact.**
> >
> >**W1: The hyperparameter ∆ used in the coarse-to-fine stage appears important to the final decision, yet its selection strategy is not discussed. There is no ablation study to show its impact on performance.**
>
> **Response to W1 and Q2 :** Thanks for your advice. Due to space limitations, we have provided a preliminary discussion on the design strategy of ∆ in **Section D.2 (lines 722–725)** of the appendix.
>
> For all datasets, we set ∆ to {0, 0.5, 1, 1.5, 2, 2.5, 3, 3.5, 4, 4.5, 5, 5.5, 6, 6.5, 7, 7.5, 8, 8.5, 9, 9.5, 10} and report the best results. For FB15k-237 and WN18RR, we set ∆ as 8 and set ∆ as 5 for NELL-995 and YAGO3-10. We additionally report the detailed model performance and its variation with different ∆, as shown in the table below.
>
> | Dataset | FB15K-237 |           |            | NELL-985  |           |            |  WN18RR  |           |            | YAGO3-10  |           |            |
> | :-----: | :-------: | :-------: | :--------: | :-------: | :-------: | :--------: | :-------: | :-------: | :--------: | :-------: | :-------: | :--------: |
> | Metrics |    MRR    | Hits@1(%) | Hits@10(%) |    MRR    | Hits@1(%) | Hits@10(%) |    MRR    | Hits@1(%) | Hits@10(%) |    MRR    | Hits@1(%) | Hits@10(%) |
> |   ∆=0   |   0.411   |   31.9    |    58.0    |   0.553   |   48.1    |    66.5    |   0.581   |   52.5    |    68.7    |   0.600   |   53.0    |    72.3    |
> |  ∆=0.5  |   0.423   |   33.0    |    59.2    |   0.572   |   50.1    |    69.3    |   0.584   |   53.0    |    68.9    |   0.613   |   54.3    |    73.0    |
> |   ∆=1   |   0.436   |   34.1    |    60.6    |   0.579   |   50.8    |    69.8    |   0.588   |   53.4    |    69.0    |   0.619   |   54.8    |    73.5    |
> |  ∆=1.5  |   0.443   |   34.7    |    61.3    |   0.585   |   51.4    |    70.5    |   0.589   |   53.6    |    69.0    |   0.623   |   55.2    |    74.0    |
> |   ∆=2   |   0.447   |   35.2    |    61.7    |   0.589   |   51.7    |    71.1    |   0.590   |   53.7    |    69.1    |   0.626   |   55.5    |    74.4    |
> |  ∆=2.5  |   0.448   |   35.3    |    61.7    |   0.590   |   51.8    |    71.2    |   0.591   |   53.8    |    69.1    |   0.628   |   55.7    |    74.6    |
> |   ∆=3   |   0.450   |   35.5    |    61.8    |   0.592   |   52.0    |    71.2    |   0.591   |   53.8    |    69.1    |   0.63    |   55.8    |    74.7    |
> |  ∆=3.5  |   0.451   |   35.6    |    61.9    |   0.592   |   52.1    |    71.2    |   0.591   |   53.8    |    69.2    |   0.631   |   56.0    |    74.8    |
> |   ∆=4   |   0.452   |   35.7    |    62.0    |   0.592   |   52.1    |    71.2    |   0.591   |   53.9    |    69.2    |   0.631   |   56.0    |    74.8    |
> |  ∆=4.5  |   0.452   |   35.7    |    62.1    |   0.592   |   52.1    |    71.2    |   0.591   |   53.9    |    69.2    |   0.631   |   56.1    |    74.9    |
> |   ∆=5   |   0.453   |   35.7    |    62.1    | **0.593** | **52.2**  |  **71.2**  |   0.592   |   53.9    |    69.2    | **0.632** | **56.1**  |  **74.9**  |
> |  ∆=5.5  |   0.453   |   35.8    |    62.2    |   0.593   |   52.2    |    71.1    |   0.592   |   53.9    |    69.2    |   0.632   |   56.1    |    74.9    |
> |   ∆=6   |   0.454   |   35.9    |    62.2    |   0.593   |   52.2    |    71.1    |   0.592   |   53.9    |    69.2    |   0.632   |   56.1    |    74.8    |
> |  ∆=6.5  |   0.454   |   35.9    |    62.3    |   0.592   |   52.2    |    71.1    |   0.592   |   53.9    |    69.2    |   0.632   |   56.1    |    74.8    |
> |   ∆=7   |   0.455   |   36.0    |    62.4    |   0.592   |   52.2    |    71.0    |   0.592   |   53.9    |    69.2    |   0.632   |   56.1    |    74.9    |
> |  ∆=7.5  |   0.456   |   36.1    |    62.5    |   0.592   |   52.2    |    71.0    |   0.593   |   54.0    |    69.9    |   0.632   |   56.1    |    74.9    |
> |   ∆=8   | **0.456** | **36.1**  |  **62.8**  |   0.592   |   52.1    |    71.0    | **0.594** | **54.2**  |  **70.0**  |   0.632   |   56.1    |    74.9    |
> |  ∆=8.5  |   0.457   |   36.1    |    62.7    |   0.592   |   52.1    |    69.9    |   0.593   |   54.0    |    69.9    |   0.632   |   56.0    |    74.9    |
> |   ∆=9   |   0.456   |   35.9    |    62.6    |   0.592   |   52.1    |    69.9    |   0.593   |   54.0    |    69.9    |   0.632   |   56.0    |    74.9    |
> |  ∆=9.5  |   0.454   |   35.9    |    62.6    |   0.592   |   52.1    |    69.9    |   0.593   |   54.0    |    69.2    |   0.632   |   56.0    |    74.9    |
> |  ∆=10   |   0.454   |   35.9    |    62.2    |   0.592   |   52.1    |    69.9    |   0.593   |   54.1    |    69.3    |   0.632   |   56.0    |    74.9    |
>
> As shown in the table, the model performance consistently improves as ∆ increases from 0, then gradually stabilizes (< 0.1% variation in MRR, Hits@1 and Hits@10) around its optimal value. Also, the model's performance is not sensitive to the value of ∆.
>
>
> >**Q3: The coarse ranking model plays a crucial role in the DuetGraph framework. Could the authors provide an ablation study or analysis to quantify how much the coarse ranking component contributes to the overall performance?**
> >
> >**W2: The contribution of the coarse ranking model to overall performance is unclear. An ablation experiment isolating the effect of modifying the coarse stage would strengthen the claims.**
>
> **Response to W2 and Q3**: Thanks for your insightful comments. The importance of this component is quantitatively demonstrated in our ablation studies presented in **Tables 3 and 4** of the manuscript.
>
> In **Table 3**, the "w/o Coarse-to-Fine reasoning" condition reveals that removing the coarse ranking component leads to performance degradation of up to **2.7%** in MRR and **2.5%** in Hits@1 metrics. This significant drop confirms the crucial role of the coarse model in our framework's overall performance.
>
> The impact of different coarse model architectures is systematically evaluated in **Table 4**. Our analysis shows that the triplet-based model emerges as the most effective choice, consistently outperforming both message passing-based and transformer-based alternatives.
>
> This finding aligns with the architectural diversity principle established in prior work [2], which suggests that **complementary model architectures yield better combined performance**. The superiority of the triplet-based approach stems from its maximal architectural difference from our fine model. While the fine model of DuetGraph specializes in capturing global information, the triplet-based coarse model focuses exclusively on local triple patterns, **maximizing architectural difference**.
>
> In contrast, the transformer-based coarse model shows weaker performance due to its **functional overlap** with our fine model's global processing. The message passing-based model takes a middle position, capturing neighborhood-level information that partially overlaps with both approaches.
>
> The results collectively show that our method performs the **best** when compared with different architectures, where the triplet model's local focus provides **the most complementary functionality** to our fine model's global view.
>
> ---
> **Reference**:
>
> [1] Balazevic et al. TuckER: Tensor Factorization for Knowledge Graph Completion. EMNLP/IJCNLP (1) 2019: 5184-5193
>
> [2] Ortega et al. Diversity and Generalization in Neural Network Ensembles. AISTATS 2022

---

> > ### Comment · Reviewer_CuGL · 2025-08-05
> >
> > Thank you for your detailed and thoughtful responses.
> >
> > Regarding Q1, the explanation on reformulating relation prediction into the tail entity completion paradigm is clear. The clarification that both tasks rely on the same scoring mechanism—fixing two elements and ranking the third—is convincing and consistent with prior approaches.
> >
> > For Q2/W1, the discussion on the hyperparameter ∆ is now sufficiently addressed. The authors provided a clear grid search strategy and included a detailed ablation analysis demonstrating the model’s robustness to different ∆ values. The empirical results show stable improvements and low sensitivity, which alleviates the initial concern.
> >
> > Regarding Q3/W2, the ablation study and architectural comparisons in Tables 3 and 4 effectively demonstrate the impact of the coarse ranking module. The quantitative degradation observed when removing the component, along with the justification for using the triplet-based architecture, provide strong support for its importance in the framework.
> >
> >
> > Overall, the author responses have resolved my concerns. I have no further questions at this point. I would like to keep my score.

---

> > > ### Author Response · Authors · 2025-08-05
> > >
> > > We are pleased that our response has successfully addressed all of your concerns. Thank you again for your detailed feedback, and for supporting acceptance!

---

### Official Review · Reviewer_GAC5 · 2025-07-03

**Clarity:** 2
**Significance:** 3
**Originality:** 2
**Rating:** 4
**Confidence:** 4

**Summary:**

This paper proposed a framework named DuetGraph for knowledge graph reasoning designed to overcome the over-smoothing problem that plagues existing methods. Duet Graph achieves this through two key techniques, dual-pathway global-local fusion and coarse-to-fine reasoning optimization. The core contributions include:
- Targets the over-smoothing issue: By decoupling and adaptively fusing information, DuetGraph mitigates the problem of incorrect answers receiving scores similar to the correct ones.
- Theoretical justification: The paper provides theoretical proofs for the effectiveness of the proposed model and optimization, providing a strong foundation for its claims.
- Performance improvement: Extensive experiments on various datasets and tasks validate the effectiveness of the proposed method.

**Questions:**

No further questions. Please refer to the weaknesses.

**Ethical Concerns:**

["NO or VERY MINOR ethics concerns only"]

**Final Justification:**

The responses clarify most of my concerns, and I decided to raise my ratings accordingly.

**Limitations:**

Yes.

**Paper Formatting Concerns:**

No.

**Quality:**

2

**Strengths And Weaknesses:**

Strengths:
- The paper targets a fundamental task and a critical issue, i.e., the over-smoothing problem.
- The paper provides theoretical proofs for the effectiveness of the proposed model and optimization.

Weaknesses:
- The presentation needs improvement. I can hardly understand the task when I first read the abstract, especially the concept of KG reasoning. It seems the task is the same with KG representation learning.
- The text description of the local and global pathways is inconsistent with Figure 1.
- In Equation (1), $\alpha$ is used to balance the local and global information. Please clarify the difference between learning the control parameter and using the graph attention network to learn different attention weights.
- In the coarse-to-fine reasoning, what is the standard of the coarse model? It seems all the baseline models could act as coarse models, and the proposed fine reasoning conducts re-ranking.
- In the evaluation, 1) What is the task definition of inductive and transductive reasoning? 2) In inductive reasoning, why divide the datasets? 3) In the transductive reasoning, the paper claims the baseline results are directly from the original papers, but there are some inconsistencies, e.g., the RotatE. If the results are reproduced based on publicly available code, please clarify where the gaps exist. 4) Section 2 claims the paper primarily focuses on the tail entity completion. Triple classification is also a typical task. Can the framework be applied to it?

---

> ### Author Rebuttal · Authors · 2025-07-31
>
> > **W1: The presentation needs improvement. I can hardly understand the task when I first read the abstract, especially the concept of KG reasoning. It seems the task is the same with KG representation learning.**
>
> **Response to W1**: Thanks for your advice. We appreciate the opportunity to clarify the relationship between KG reasoning and representation learning, as well as to improve the presentation of the abstract.
>
> ***Clarifying KG Reasoning vs. Representation Learning***. While the concepts of the two are closely related, they serve for different purposes:
>
> - **KG Representation Learning** encodes KG entities and relations into vector representations, capturing their semantic and structural relationships. This enables computational understanding and enhances KG's expressiveness.
>
> - **KG Reasoning** focuses on inferring missing information (e.g., predicting new facts) from existing data, thus addressing KG incompleteness. As noted in Section 1 of [1], reasoning relies on the embeddings learned through representation learning to perform tasks such as link prediction.
>
> **In summary**, KG representation learning provides the foundational embeddings, while reasoning uses them for knowledge inference [2].
>
> **Definitions in the Manuscript**. We define KG reasoning explicitly in **lines 21-22 (Section 1)** and **lines 81-86 (Section 2)**, following the way of concept formulation in prior work, e.g., **Section 3 of [3]**, **Section 2 of [4]**, and **Section 1 of [1]**.
>
> **Transferability of Learned Representations**. Beyond reasoning tasks, DuetGraph's learned KG representations demonstrate strong transferability, achieving SOTA performance on downstream tasks like triplet classification (see Response to W5(4) for experimental results).
>
> Following your advice, we will revise the abstract to clarify this definition of KG reasoning and differentiate it from KG representation learning.
>
>
> > **W2: The text description of the local and global pathways is inconsistent with Figure 1.**
>
> **Response to W2**: You are right. There are two numbering errors in the textual descriptions of the pathways (**lines 112 and 113**):
>
> -  "local pathway (Step 2)" should be "local pathway (Step 3)"
> -  "global pathway (Step 3)" should be "global pathway (Step 2)"
>
> We apologize for any confusion this may have caused. These corrections will be made in the revised manuscript
>
> > **W3: In Equation (1), α is used to balance the local and global information. Please clarify the difference between learning the control parameter and using the graph attention network to learn different attention weights.**
>
> **Response to W3**: Thank you for your insightful question regarding the distinction between learning the control parameter α and using graph attention weights. We clarify this difference from three perspectives:
>
> - **Technical Role**. As illustrated in Figure 2, the graph attention mechanism (Step 2) first learns global weights, which subsequently inform the learning of the control parameter α (Step 4). The attention weights serve as intermediate representations that enable α to effectively balance local and global information.
>
> - **Theoretical Advantage**. Introducing α allows for better fusion of the global weights captured by the attention mechanism (Step 2) and the local weights acquired via message passing (Step 3), which helps mitigate over-smoothing and enhances the model’s quality, as shown in **Theorem 1** of the manuscript.
>
> - **Experimental Study**. Incorporating α achieves better performance compared to using attention mechanism alone, as shown in the table below.
>
> |Dataset| FB15k-237|||WN18RR|||NELL-995|||YAGO3-10|||
> |:-:|:-:|:-:|:-:|:-:|:-:|:-:|:-:|:-:|:-:|:-:|:-:|:-:|
> |Metrics|MRR|Hits@1|Hits@10|MRR|Hits@1|Hits@10|MRR|Hits@1|Hits@10|MRR|Hits@1| Hits@10|
> |w/ α|**0.456**|**36.1**|**62.8**|**0.594**|**54.2**| **70.0**|**0.593**|**52.2**|**71.2**|**0.632**|**56.1**| **74.9**|
> |only w/ attention|0.445|35.1|61.2|0.584|54.1|69.0|0.582|51.0| 70.3|0.617|54.2|73.7|
>
>
> ​
> > **W4: In the coarse-to-fine reasoning, what is the standard of the coarse model? It seems all the baseline models could act as coarse models, and the proposed fine reasoning conducts re-ranking.**
>
> **Response to W4**: Thanks for your insightful comments.
>
> **Architectural Diversity Principle.** As established in prior work [5], maximizing architectural diversity between coarse and fine models leads to better overall performance. This principle also applies for coarse model selection in DuetGraph.
>
> **Performance Validation.** Our experiments in **Table 4** evaluate different coarse model categories (triplet-based, message passing-based, and transformer-based) and their effect on final model quality, where triplet-based model focuses exclusively on local triple-level patterns, message passing model captures neighborhood-level information, and transformer model handles global patterns (like our fine model).
>
> As demonstrated in **Table 4**, the triplet-based model achieves the best performance as a coarse model due to its maximal architectural difference from our global-information-focused fine model. This supports the diversity principle.
>
>
>
> >**W5: In the evaluation,**
> >(1) **What is the task definition of inductive and transductive reasoning?**
> >(2) **In inductive reasoning, why divide the datasets?**
> >(3) **In the transductive reasoning, the paper claims the baseline results are directly from the original papers, but there are some inconsistencies, e.g., the RotatE. If the results are reproduced based on publicly available code, please clarify where the gaps exist.**
> >(4) **Section 2 claims the paper primarily focuses on the tail entity completion. Triple classification is also a typical task. Can the framework be applied to it?**
>
> **Response to W5**: Thanks for your valuable comments.
>
> (1) **Transductive vs. Inductive Reasoning:** Following the formal definition in [6], transductive reasoning assumes that all test entities appear during training, while inductive reasoning handles completely unseen entities during testing. This difference is fundamental to evaluating model generalization capabilities.
>
> (2) **Dataset Construction for Inductive Evaluation:** Following the methodology in [7], we construct our inductive evaluation datasets by ensuring complete separation between training and test entities. This strict partitioning, where test entities are excluded from training, enables a reliable assessment of the model's generalization capability to unseen knowledge.
>
> (3)  **Reproduction of RotatE Results:** We clarify that the results of RotatE are reproduced based on publicly available code in [8]. For more rigorous evaluations, we adopted the widely used data augmentation strategy in [9, 10, 11] by incorporating inverse triplets [12] during both training and inference, because this can ensure symmetry and coherence between queries and answers, as shown in [13]. Notably, this differs from the original RotatE implementation [8], which excluded inverse triplets. This accounts for the performance variation observed.
>
> (4)  **Performance on Triple Classification:** To demonstrate DuetGraph's generalizability, we evaluate its performance on triple classification using the following setup:
>
> - Datasets: UMLS [14], FB13 [15] and WN11 [15] (Standard benchmark datasets for knowledge graph tasks)
> - Baselines: triplet-based (HousE [16]), message passing-based (AdaProp [17]), transformer-based (HittER [18]), and hybrid message passing-transformer (KnowFormer [3]) models (SOTA methods for comprehensive comparison)
>
> |Method|UMLS Accuracy (%)|FB13 Accuracy (%)|WN11 Accuracy (%)|
> |:-:|:-:|:-:|:-:|
> |HousE|83.1|69.8|65.3|
> |HittER|59.4|62.2|69.6|
> |AdaProp|77.0|71.9|67.1|
> |KnowFormer|83.1|77.3|70.2|
> |DuetGraph|**84.3**|**80.0**|**71.9**|
>
> The experimental results demonstrate that DuetGraph consistently outperforms all baseline methods across all datasets, achieving new SOTA performance on the triple classification task.
>
> ---
> **Reference:**
>
> [1] Wu et al. KRLGI: Knowledge Representation Learning Based on Global Information for Reasoning. ICTAI 2024
>
> [2] Chen et al. An Overview of Knowledge Graph Reasoning: Key Technologies and Applications. J. Sens. Actuator Networks 2022
>
> [3] Liu et al. KnowFormer: Revisiting Transformers for Knowledge Graph Reasoning. ICML 2024
>
> [4] Zhu et al. A\*Net: A Scalable Path-based Reasoning Approach for Knowledge Graphs. NeurIPS 2023
>
> [5] Ortega et al. Diversity and Generalization in Neural Network Ensembles. AISTATS 2022
>
> [6] Zhang et al. Logical Reasoning with Relation Network for Inductive Knowledge Graph Completion. KDD 2024
>
> [7] Chen et al. Meta-Knowledge Transfer for Inductive Knowledge Graph Embedding. SIGIR 2022
>
> [8] Sun et al. RotatE: Knowledge Graph Embedding by Relational Rotation in Complex Space. ICLR 2019
>
> [9] Kazemi et al. SimplE Embedding for Link Prediction in Knowledge Graphs. NeurIPS 2018
>
> [10] Wang et al. SimKGC: Simple Contrastive Knowledge Graph Completion with Pre-trained Language Models. ACL 2022
>
> [11] Xiong et al. DeepPath: A Reinforcement Learning Method for Knowledge Graph Reasoning. EMNLP 2017
>
> [12] Lacroix et al. Canonical Tensor Decomposition for Knowledge Base Completion. ICML 2018
>
> [13] Li et al.  KERMIT: Knowledge graph completion of enhanced relation modeling with inverse transformation. Knowl. Based Syst. 2025
>
> [14] Olivier et al. The Unified Medical Language System (UMLS): integrating biomedical terminology. Nucleic Acids Res. 2004
>
> [15] Socher et al. Reasoning With Neural Tensor Networks for Knowledge Base Completion. NIPS 2013
>
> [16] Li et al. HousE: Knowledge Graph Embedding with Householder Parameterization. ICML 2022
>
> [17] Zhang et al. AdaProp: Learning Adaptive Propagation for Graph Neural Network based Knowledge Graph Reasoning. KDD 2023
>
> [18] Chen et al. HittER: Hierarchical Transformers for Knowledge Graph Embeddings. EMNLP 2021

---

### Official Review · Reviewer_qRnh · 2025-07-10

**Clarity:** 3
**Significance:** 3
**Originality:** 3
**Rating:** 5
**Confidence:** 2

**Summary:**

This paper introduces DuetGraph, a novel knowledge graph (KG) reasoning framework designed to mitigate the problem of "score over-smoothing"  which often blurs the distinction between correct and incorrect answers in existing KG reasoning methods. DuetGraph addresses this through two main innovations: a dual-pathway global-local fusion mechanism and a coarse-to-fine reasoning optimization.

**Questions:**

Adaptive Fusion Parameter (α): The paper mentions that the adaptive fusion approach "drives α below the theoretical threshold in Theorem 1 via parameter update with gradient descent". Could the authors elaborate on how this parameter α is initialized and empirically how its value changes during training across different datasets? Are there any cases where α does not converge as expected, and what might be the implications?


Impact of Coarse Model Choice: Table 4 shows DuetGraph's performance with different coarse models. While the results are consistently good, can the authors provide more insight into the characteristics of a "good" coarse model for DuetGraph? For instance, does the performance of the coarse model directly correlate with the final DuetGraph performance, or does DuetGraph's coarse-to-fine mechanism mitigate weaknesses of less effective coarse models?

**Ethical Concerns:**

["NO or VERY MINOR ethics concerns only"]

**Final Justification:**

agree to accept

**Limitations:**

Yes

**Quality:**

3

**Strengths And Weaknesses:**

Originality: The core idea of segregating local and global information processing into dual pathways, rather than stacking them, is novel and directly addresses the identified problem of score over-smoothing. The coarse-to-fine optimization further enhances this by refining the candidate space

Generalizability of Coarse Models: While DuetGraph shows consistent outperformance with various coarse-grained models, deeper analysis into why certain coarse models might perform better or worse within the DuetGraph framework could provide more insight.

---

> ### Author Rebuttal · Authors · 2025-07-31
>
> >**Q1: Adaptive Fusion Parameter (α): The paper mentions that the adaptive fusion approach "drives α below the theoretical threshold in Theorem 1 via parameter update with gradient descent". Could the authors elaborate on how this parameter α is initialized and empirically how its value changes during training across different datasets? Are there any cases where α does not converge as expected, and what might be the implications?**
>
> **Response to Q1**: Thanks for your comments. In our setting, we initialize the value of α randomly within the range (0, 1). Since we use the same random seed for all datasets, the initial value of α is identical across different datasets and is 0.549, as shown in the table below. Following your advice, we report the range of α values observed during training across different datasets, as shown in the table below.
>
> | Datasets | FB15k-237_v1 | FB15k-237_v2 | FB15k-237_v3 | FB15k-237_v4 |
> |:--------:|:------------:|:------------:|:------------:|:------------:|
> | Theoretical threshold |     2.27     |     2.44     |     2.40     |      2.46    |
> | α (Epoch=0) |     0.549    |     0.549    |    0.549    |     0.549    |
> | α (Epoch=2) |     0.538    |     0.518    |    0.513    |     0.507    |
> | α (Epoch=4) |     0.531    |     0.501    |    0.498    |     0.486    |
> | α (Epoch=6) |     0.522    |     0.485    |    0.477    |     0.465    |
> | α (Epoch=8) |     0.514    |     0.468    |    0.460    |     0.450    |
> | α (Epoch=10) |     0.508    |     0.457    |    0.445    |     0.437    |
> | α (Epoch=12) |     0.509    |     0.458    |    0.445    |     0.439    |
> | α (Epoch=14) |     0.511    |     0.460    |    0.447    |     0.440    |
> | α (Epoch=16) |     0.512    |     0.461    |    0.448    |     0.441    |
> | α (Epoch=18) |     0.512    |     0.461    |    0.449    |     0.441    |
> | α (Epoch=20) |     0.512    |     0.461    |    0.449    |    0.441    |
>
> | Datasets | NELL-995_v1 | NELL-995_v2 | NELL-995_v3 | NELL-995_v4 |
> |:--------:|:------------:|:------------:|:------------:|:------------:|
> | Theoretical threshold | 2.49    | 2.58    | 2.54    | 2.42    |
> | α (Epoch=0) | 0.549   | 0.549   | 0.549   | 0.549   |
> | α (Epoch=2) | 0.544   | 0.536   | 0.526   | 0.536   |
> | α (Epoch=4) | 0.539   | 0.532   | 0.496   | 0.516   |
> | α (Epoch=6) | 0.523   | 0.525   | 0.474   | 0.500   |
> | α (Epoch=8) | 0.514   | 0.517   | 0.447   | 0.487   |
> | α (Epoch=10) | 0.514   | 0.509   | 0.424   | 0.477   |
> | α (Epoch=12) | 0.514   | 0.509   | 0.425   | 0.475   |
> | α (Epoch=14) | 0.512   | 0.510   | 0.425   | 0.476   |
> | α (Epoch=16) | 0.512   | 0.510   | 0.426   | 0.476   |
> | α (Epoch=18) | 0.512   | 0.510   | 0.427   | 0.476   |
> | α (Epoch=20) | 0.512   | 0.510   | 0.427   | 0.476   |
>
> |       Datasets        | WN18RR_v1 | WN18RR_v2 | WN18RR_v3 | WN18RR_v4 |
> | :-------------------: | :-------: | :-------: | :-------: | :-------: |
> | Theoretical threshold |   2.17    |   2.00    |   2.03    |   2.10    |
> |      α (Epoch=0)      |   0.549   |   0.549   |   0.549   |   0.549   |
> |      α (Epoch=2)      |   0.539   |   0.543   |   0.546   |   0.543   |
> |      α (Epoch=4)      |   0.534   |   0.531   |   0.534   |   0.540   |
> |      α (Epoch=6)      |   0.533   |   0.524   |   0.536   |   0.537   |
> |      α (Epoch=8)      |   0.532   |   0.512   |   0.523   |   0.536   |
> |     α (Epoch=10)      |   0.531   |   0.504   |   0.511   |   0.532   |
> |     α (Epoch=12)      |   0.531   |   0.505   |   0.511   |   0.532   |
> |     α (Epoch=14)      |   0.532   |   0.505   |   0.510   |   0.533   |
> |     α (Epoch=16)      |   0.533   |   0.505   |   0.511   |   0.533   |
> |     α (Epoch=18)      |   0.533   |   0.505   |   0.511   |   0.533   |
> |     α (Epoch=20)      |   0.533   |   0.505   |   0.511   |   0.533   |
>
> The results show that α consistently converges to a stable value during training, with negligible fluctuations afterward (less than 0.001). Across all datasets, α converges reliably as expected. Moreover, α remains below the theoretical threshold
>
> $\frac{\sigma_{max}(\mathcal{M}_{\mathrm{D}})+\sigma\_{max}(\mathcal{M}\_{\mathrm{O}})}{1+\sigma\_{max}(\mathcal{M}\_{\mathrm{O}})}$
>
> as stated in **Theorem 1** of the manuscript.
> This supports our claim that the proposed dual-pathway model more effectively mitigates score over-smoothing than the single-pathway model, aligning with the theoretical bound.
>
> >**Q2: Impact of Coarse Model Choice: Table 4 shows DuetGraph's performance with different coarse models. While the results are consistently good, can the authors provide more insight into the characteristics of a "good" coarse model for DuetGraph? For instance, does the performance of the coarse model directly correlate with the final DuetGraph performance, or does DuetGraph's coarse-to-fine mechanism mitigate weaknesses of less effective coarse models?**
> >
> >**W1: Generalizability of Coarse Models: While DuetGraph shows consistent outperformance with various coarse-grained models, deeper analysis into why certain coarse models might perform better or worse within the DuetGraph framework could provide more insight.**
>
> **Response to Q2 and W1**:
> Thanks for your insightful comments.
>
> ***Key Observations from Table 4.*** Our experiments in **Table 4** evaluate different coarse model categories (triplet-based, message passing-based, and transformer-based) and their effect on final model quality.
> The results show: triplet-based models perform best as coarse models; message passing-based models rank second; and transformer-based models yield the weakest performance in this role.
>
> ***Characteristics of a Good Coarse Model.***
> As supported by prior work [1], **architectural diversity between coarse and fine models enhances overall performance**.
> Specifically, the fine model of DuetGraph captures global information, while the triplet-based coarse model focuses only on local triple-level patterns, maximizing architectural difference. In contrast, transformer-based coarse model also handles global information, making its architectural gap from the fine model the smallest. The message passing-based model captures information from local neighborhoods and offers a moderate diversity. This explains why the triplet-based model—despite its simplicity—emerges as the most effective coarse model.
>
> ***Does Coarse Model Performance Directly Affect DuetGraph?*** The answer is **NO**. As shown in **Tables 2** and **4** of the manuscript, the performance of DuetGraph does not depend on the standalone performance of the coarse model. Instead, it compensates for weaker coarse models by refining their outputs. For example, when integrating the **worst** triplet-based model (e.g., TransE [2]) into the coarse model in dataset WN18RR, DuetGraph boosts its **MRR** (Model Quality Metric) from **0.222** to **0.593**, **Hits@1** from **1.4%** to **54.2%**, and **Hits@10** from **52.8%** to **69.0%**, resulting in significant improvements of **0.371**, **52.8%**, and **16.2%**, respectively. Moreover, the enhanced performance of TransE [2] exceeds that of the SOTA model (e.g., KnowFormer [3]), with respective **MRR**, **Hits@1**, and **Hits@10** values of **0.579**, **52.8%**, and **68.7%**, as presented in **Table 2** of the manuscript.
>
> We will include the analysis in the revised version of the manuscript to clarify these points.
>
> ---
> **Reference**:
>
> [1] Ortega et al. Diversity and Generalization in Neural Network Ensembles. AISTATS 202
>
> [2] Bordes et al. Translating Embeddings for Modeling Multi-relational Data. NIPS 2013: 2787-2795
>
> [3] Liu et al. KnowFormer: Revisiting Transformers for Knowledge Graph Reasoning. ICML 2024

---

### Note · Authors · 2025-08-12

Dear NeurIPS 2025 **Reviewers, ACs and SACs**,

Thank you for handling our submission.

We would like to express our gratitude to **all reviewers** for their valuable feedback and are especially grateful to **Reviewers qRnh, CuGL and hDcp**. They affirmed our work’s **novelty**, **rigorous theoretical analysis**, and **solid empirical support**.

While we appreciate **Reviewer GAC5’s** acknowledgment that our work provides strong theoretical proofs for the effectiveness and targets a fundamental task and a critical issue, **Reviewer GAC5’s** borderline rejection rating is rooted in **an insufficient understanding of our work**, which we believe is not an appropriate basis for judging our contribution.

We addressed **Reviewer GAC5’s** concerns from these three perspectives:

- **Fundamental Concepts.** We have already defined the KG concepts in the manuscript **(lines 21-22, Section 1; lines 81-86, Section 2)**, and these concepts have been established in prior works. In our rebuttal, we further cited prior works to highlight and clarify these concepts.
- **Ablation Study.** In the rebuttal, we provided a further explanation of our ablation study results, which is consistent with our response to **Reviewer CuGL (Q3)**, who has confirmed that our response addressed the concerns.
- **Generalization.** In rebuttal, we conducted additional experiments to confirm that our framework applies to other tasks **(e.g., triple classification task)**.

Overall, we believe that **Reviewer GAC5’s** concerns regarding the **fundamental concepts**, **ablation study**, and **generalization** stem from an insufficient understanding of our work and **we have addressed all the concerns during rebuttal**. Moreover, **Reviewer GAC5** did not participate in discussions with us during the rebuttal period. Therefore, we would appreciate it if you could take this situation into consideration when making the final decision.

Thank you for your hard work and support.

Best regards,

The authors of Paper **DuetGraph: Coarse-to-Fine Knowledge Graph Reasoning with Dual-Pathway Global-Local Fusion**

---

### Decision · Program_Chairs · 2025-09-17

**Decision:**

Accept (poster)

**Comment:**

This paper presents DuetGraph, a novel dual-pathway, coarse-to-fine framework for knowledge graph reasoning that effectively mitigates the over-smoothing problem. The architecture is well-motivated, supported by theoretical analysis, and demonstrates state-of-the-art performance across a wide range of benchmarks.

The authors' exemplary engagement during the rebuttal period was pivotal. They comprehensively addressed all reviewer concerns by conducting significant new experiments. Key initial weaknesses regarding efficiency, scalability, and evaluation rigor were transformed into strengths. Notably, the authors:
1) Validated DuetGraph's efficiency and scalability on very large KGs (Wikidata5M, Freebase), demonstrating superior performance over challenging baselines.

2) Re-ran all evaluations using a stricter ranking protocol suggested by a reviewer, confirming the robustness of their state-of-the-art claims.

In summary, the authors addressed every point raised with new experiments and detailed analysis. While Reviewer hDcp adjusted some sub-scores downwards based on a revised perspective of the work's novelty, they acknowledged that all their technical questions were answered and maintained their overall positive recommendation. This extensive rebuttal was pivotal in solidifying the paper's contribution and justifying its acceptance.